# MYC promotes immune-suppression in triple-negative breast cancer via inhibition of interferon signaling

Dario Zimmerli[1,2,14], Chiara S. Brambillasca[1,2,14], Francien Talens[3,14], Jinhyuk Bhin [1,2,4], Renske Linstra [3], Lou Romanens[1,5], Arkajyoti Bhattacharya [3], Stacey E. P. Joosten[2,6], Ana Moises Da Silva[1,2], Nuno Padrao[2,6], Max D. Wellenstein[2,7,11], Kelly Kersten [6,12], Mart de Boo[1], Maurits Roorda [3], Linda Henneman[8], Roebi de Bruijn[1,2,4], Stefano Annunziato[1,2], Eline van der Burg[1,2], Anne Paulien Drenth[1,2], Catrin Lutz[1,2], Theresa Endres [9,13], Marieke van de Ven[10], Martin Eilers [9], Lodewyk Wessels [2,4], Karin E. de Visser [2,7], Wilbert Zwart [2,6], Rudolf S. N. Fehrmann [3] ✉, Marcel A. T. M. van Vugt [3] ✉ & Jos Jonkers [1,2] ✉

The limited efficacy of immune checkpoint inhibitor treatment in triple-negative breast cancer (TNBC) patients is attributed to sparse or unresponsive tumor-infiltrating lymphocytes, but the mechanisms that lead to a therapy resistant tumor immune microenvironment are incompletely known. Here we show a strong correlation between MYC expression and loss of immune signatures in human TNBC. In mouse models of TNBC proficient or deficient of breast cancer type 1 susceptibility gene (BRCA1), MYC overexpression dramatically decreases lymphocyte infiltration in tumors, along with immune signature remodelling. MYC-mediated suppression of inflammatory signalling induced by BRCA1/2 inactivation is confirmed in human TNBC cell lines. Moreover, MYC overexpression prevents the recruitment and activation of lymphocytes in both human and mouse TNBC co-culture models. Chromatin-immunoprecipitation-sequencing reveals that MYC, together with its co-repressor MIZ1, directly binds promoters of multiple interferon-signalling genes, resulting in their downregulation. MYC overexpression thus counters tumor growth inhibition by a Stimulator of Interferon Genes (STING) agonist via suppressing induction of interferon signalling. Together, our data reveal that MYC suppresses innate immunity and facilitates tumor immune escape, explaining the poor immunogenicity of MYC-overexpressing TNBCs.

Breast cancer is among the leading causes of cancer-associated death in women, with a lifetime risk of ~12.5%[1]. Triple-negative breast cancers (TNBC) lack expression of the ER, PR and HER2 receptors. A substantial fraction of TNBCs is defective in DNA repair via homologous recombination (HR) due to genetic or epigenetic inactivation of BRCA1 or other components of the HR pathway[2]. Although TNBC only represents 15–20% of breast carcinomas, distant recurrence and mortality in TNBC are significantly higher when compared to other breast cancer subtypes[3]. As very few targeted therapeutic options are available for patients with TNBC, radiation and chemotherapy are the current

standard-of-care treatments, prompting the need for new and more effective treatments[4].

Targeting the immune system is increasingly employed as a successful treatment approach for cancer. Immune checkpoint inhibitors (ICI) have resulted in survival benefits across multiple tumor types, with high mutational load and tumor-infiltrating lymphocytes (TILs) being associated with response[5]. TNBCs were also reported to have high levels of TILs[6,7], which was shown to be predictive of treatment response to conventional chemotherapeutics[8,9]. Unfortunately, clinical evaluation of single-agent ICI therapy in patients with TNBC only showed benefit in the minority of cases[10,11]. The poor efficacy of ICI in TNBC patients is surprising, because TNBCs are characterized by multiple features that are associated with response to ICI, including high levels of TILs and potentially high levels of neo-epitopes due to their frequent DNA repair defects, which cause pronounced copy number aberrations and complex rearrangements[12].

Recently, inactivation of BRCA1/2 and the ensuing DNA damage was shown to result in the accumulation of DNA in the cytosol and subsequent activation of the *cyclic GMP-AMP synthase* (cGAS) / *stimulator of interferon (IFN) genes* (STING) pathway[13,14]. Originally discovered as an anti-viral pathway responding to non-self DNA in the cytosol[15], the cGAS/STING pathway was recently described to also respond to 'own' DNA, when outside the nucleus[16,17]. Interestingly, this innate immune pathway was demonstrated to be required for a robust adaptive anti-tumor immune response[18,19]. Apparently, TNBCs, and *BRCA1* mutant tumors in particular, have evolved mechanisms to suppress immune responses induced by neo-antigen expression and cGAS/STING signaling.

Multiple recurring gene alterations in tumor-suppressor genes and oncogenes have been described for TNBC. For instance, mutations in the tumor suppressor *TP53* are commonly found along with *BRCA1* in human TNBC[20]. Also, the transcription factor *MYC*, which resides in the 8q24 locus, is regularly amplified in TNBC and especially in *BRCA1*-mutated TNBC[20]. In line with this, a transcriptional signature associated with *MYC* amplification is correlated with a gene signature of *BRCA1*-deficient breast cancers[21]. MYC regulates global gene expression and thus promotes proliferation as well as many other cellular processes[22,23]. Interestingly, MYC was shown to not only promote transcription of targets, but depending on the associated co-factors can also repress transcription[24]. Notably, recent studies have shown that MYC influences the host tumor microenvironment and immune effectors in liver, lung and pancreatic cancer[25–28], suggesting a role for MYC in immune suppression beyond its activity as a mitogen. Interestingly, immunogenomic analysis of human TNBC provided hints that MYC overexpression correlates with low immune infiltration (Xiao et al.[29]). However, functional proof that MYC regulates the immune system in TNBC is still lacking and the potential underlying mechanisms are largely unknown.

Here, we explore whether MYC might directly influence immune evasion in TNBC, using a mouse model that recapitulates key features of TNBC. We show that MYC suppresses STING-IFN signaling in a tumor cell-intrinsic fashion via direct transcriptional repression, thereby blunting immune cell invasion in TNBC. Boosting immune infiltration in MYC-expressing TNBCs via direct enhancement of interferon signaling should therefore be a promising avenue to improve ICI therapies in such tumors.

## Results

### MYC expression associates with downregulation of inflammatory pathways in human breast cancer

In TNBCs, the most frequently aberrated oncogene is *MYC*, which was found to be amplified in 61.27% of all TNBC samples within the The Cancer Genome Atlas (TCGA) database (Fig. 1A). *MYC* is also the most commonly amplified oncogene in *BRCA1/2*-mutated breast cancers (Supplementary Fig. S1A)[20]. To assess the impact of *MYC* expression on

inflammatory signaling in TNBCs, we performed gene set enrichment analysis (GSEA) on RNA sequencing (RNA-seq) data obtained from pre-treatment tumor samples from the TONIC phase II trial (NCT02499367), which evaluated the efficacy of nivolumab after immune induction in TNBC patients (Voorwerk et al.[30]). As expected, *MYC* expression positively correlated with MYC target gene sets and E2F targets (Fig. 1B). Interestingly, *MYC* expression negatively correlated with IFN and JAK-STAT signaling, as well as other inflammatory pathways, including 'IL2-STAT5 signaling', 'allograft rejection' and 'complement activation' (Fig. 1B, C). This is especially interesting considering the fact that in the TONIC trial, increased TILs and immune scores were associated with higher response rates to ICI treatment (Voorwerk et al.[30]). In an independent TCGA dataset, analysis was performed on gene expression data of breast cancer samples that were stratified for amplification or mutation status of selected oncogenes (Supplementary Fig. S1A, B). *MYC* amplification again correlated with suppression of gene sets related to inflammatory signaling (Supplementary Fig. S1C). Of note, also other oncogenes seemed to correlate with reduced IFN and JAK-STAT signaling (Supplementary Fig. S1C). However, most oncogene amplifications co-occurred with *MYC* amplification (Fig. S1A) or led to upregulated MYC transcriptional signatures, indicating that these effects may also directly or indirectly reflect effects of MYC.

To confirm the effects of MYC overexpression on transcriptional re-wiring in an experimental setting, the TNBC cell lines BT-549, MDA-MB-231 and HCC1806 were engineered to overexpress MYC in a doxycycline (dox)-inducible manner. Next, gene expression was analyzed using RNA-seq. To investigate the biological processes that are affected by oncogene expression in TNBC cells, GSEA of the gene expression data was performed. Upon MYC overexpression, a strong suppression of IFN signaling pathways was observed, confirming our findings from the human patient samples (Fig. 1D). Taken together, these findings suggest that MYC overexpression suppresses inflammatory signaling, supporting breast cancer to evade detection by the immune system.

### MYC-overexpressing mouse TNBCs display an immune-depleted microenvironment

To explore if MYC regulates immune responses in mammary tumors in vivo, we used four genetically engineered mouse models (GEMM) of BRCA1-proficient and -deficient TBNC with or without engineered MYC overexpression: *WapCre;Trp53^{F/F}* (WP), *WapCre;Trp53^{F/F};Col1a1^{invCAG-Myc-IRES-Luc/+}* (WP-Myc), *WapCre;Brca1^{F/F};Trp53^{F/F}* (WB1P) and *WapCre;Brca1^{F/F};Trp53^{F/F};Col1a1^{invCAG-Myc-IRES-Luc/+}* (WB1P-Myc)[20]. GSEA of RNA-seq data of mammary tumors of a size of 1500 mm³ from these four GEMMs showed a clear reduction of immune signatures in the WP-Myc and WB1P-Myc tumors with engineered MYC overexpression, when compared to WP and WB1P control tumors (Fig. 2A). Consistently, unsupervised hierarchical clustering of all tumors based on expression of IFN-stimulated genes (ISGs) (Saleiro et al.[31]) resulted in clustering according to MYC status (Supplementary Fig. S2A). Indeed, WB1P and WP models showed similar change of gene expression profiles upon MYC expression, indicating the dominant role of MYC in shaping the transcriptional landscape (R = 0.67, P < 2.2 × 10^{-16}) (Supplementary Fig. S2B). Expression of previously published MYC signature genes was consistently increased in our WB1P-Myc versus WB1P models[32], confirming the functionality of our MYC-overexpressing mouse models (Supplementary Fig. S2C). Importantly, immunohistochemical analysis of the same tumors showed a significant reduction of tumor-infiltrating lymphocytes (TIL) in WP-Myc and WB1P-Myc tumors compared to WP and WB1P tumors (Fig. 2B, C).

Since the effect of MYC overexpression on CD3⁺ TILs was most profound in the BRCA1-deficient mammary tumors (Fig. 2B), we further focused on the WB1P model. In line with our transcriptomic and histopathologic analysis, flow cytometry analysis of immune cell

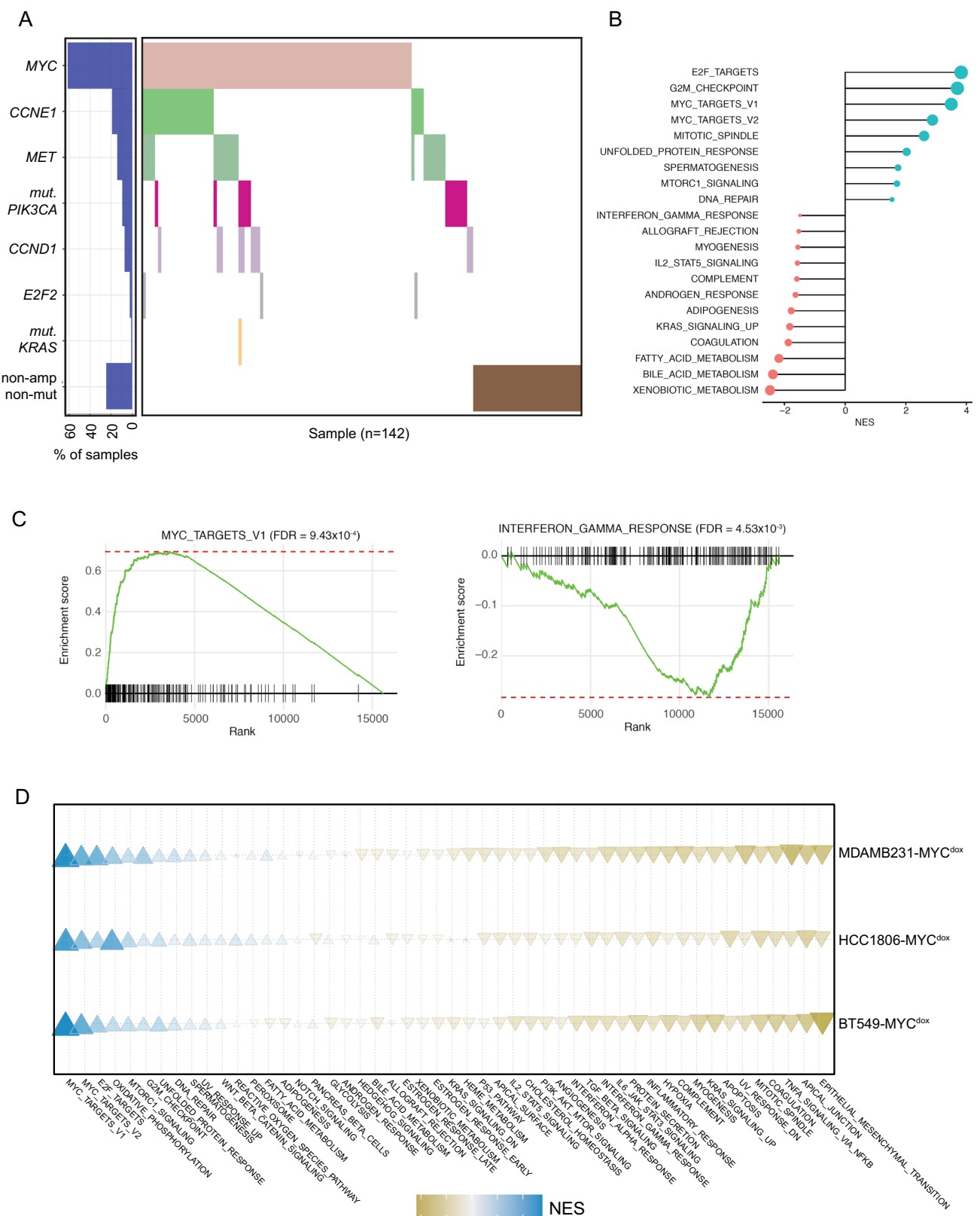

**Fig. 1 | MYC expression is associated with downregulation of inflammatory pathways in human breast cancer. A** Distribution plot of TNBC samples used for GSEA analysis from TCGA data. In total, 142 samples were included in the analyses. Individual samples are plotted on the x-axis. **B** GSEA for the genes which are positively (cyan) or negatively (pink) correlated with expression of MYC in the TONIC trial dataset. The normalized enrichment scores (NES) for the significantly enriched gene sets (FDR < 0.05) are presented in the bar plot. MsigDB Hallmark gene sets were used for GSEA analysis. **C** GSEA plots for two significant gene sets, MYC_TARGETS_V1 and INTERFERON_GAMMA_RESPONSE. **D** GSEA of TNBC cell lines BT-549, MB-231 and HCC1806 overexpressing MYC in a doxycycline-inducible manner was performed. Depicted is a bubble plot of the enrichment of specific gene sets between the three tested cell lines upon MYC induction. Increased expression is depicted in blue, while repression is depicted in orange.

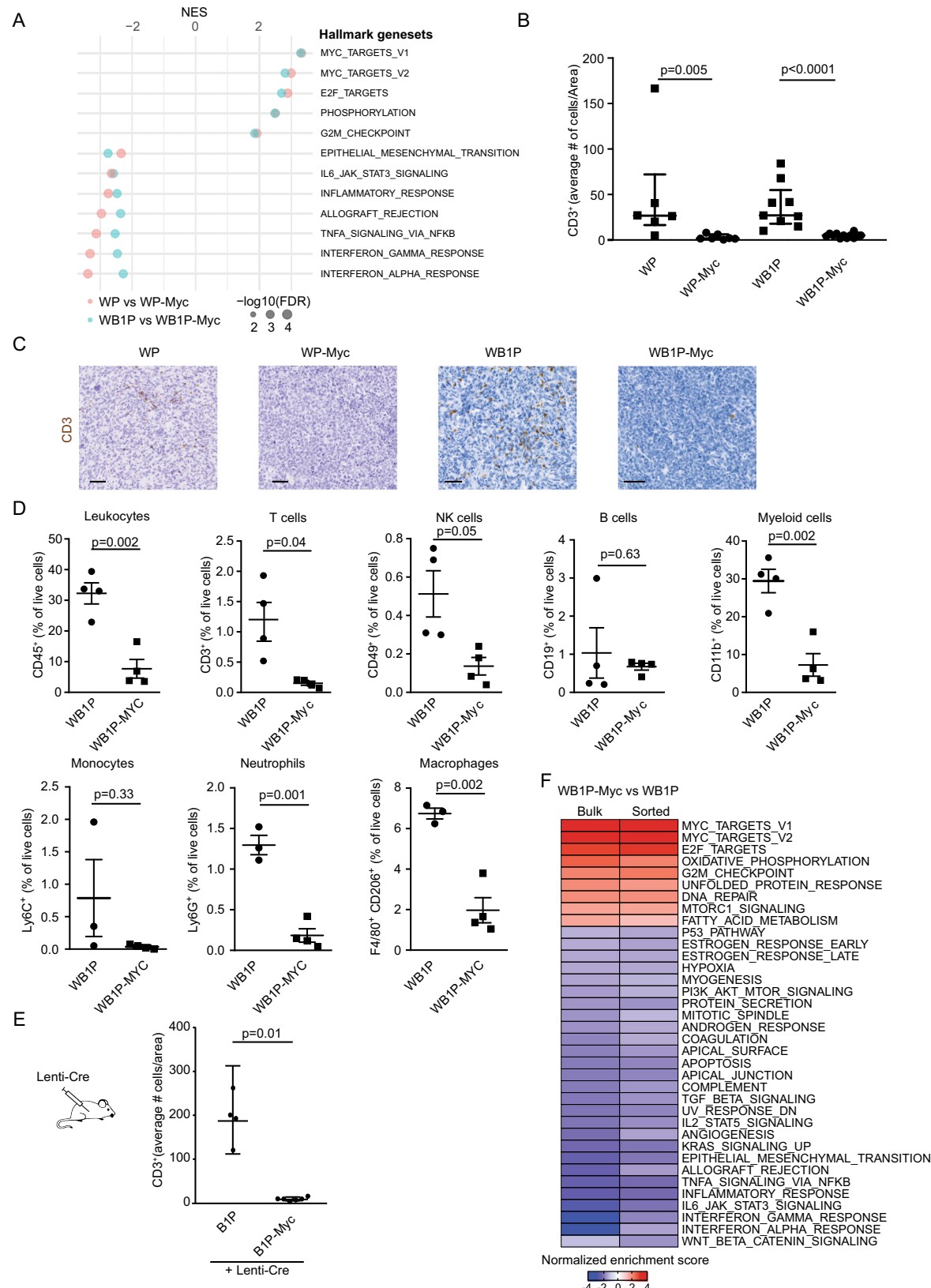

populations in WB1P-Myc versus WB1P tumors showed a clear loss of CD3$^+$ T cells and decreased frequencies of infiltrating CD49b$^+$ NK cells and CD11b$^+$ myeloid cells including Neutrophils (Ly6C$^+$) and Macrophages (F4/80$^+$, CD206$^+$) in WB1P-Myc tumors (Fig. 2D, Supplementary Fig. S2E). In contrast, we did not observe a significant difference in CD19$^+$ B cell frequencies (Fig. 2D). Of note, draining lymph nodes,

spleen, and blood showed similar lymphocyte frequencies in WB1P and WB1P-Myc mice, arguing against systemic immune-suppression and pointing towards local dampening of the immune response via paracrine signals from tumor cells (Supplementary Fig. S2D, E). To further corroborate our findings from the WB1P and WB1P-Myc tumor models, we used somatic engineering[33] to induce mammary tumors in

**Fig. 2 | MYC-overexpressing mouse TNBCs display an immune-depleted microenvironment. A** MSigDB Hallmark gene sets significantly represented by WP vs WP-Myc (pink) and WB1P vs WB1P-Myc (cyan) tumors from GSEA analysis. The normalized enrichment scores (NES) for the significantly enriched gene sets (FDR < 0.05) are presented in the bar plot. **B** Quantification of CD3[+] T cells in WP (*n* = 6), WP-Myc (*n* = 7), WB1P (*n* = 9) and WB1P-Myc (*n* = 12) tumors (unpaired 2-sided Mann-Whitney test, *p* = 0.005 for WP vs WP-Myc and *p* < 0.0001 for WB1P vs WB1P-Myc, mean with SD is plotted). **C** Representative immunostainings (of at least 5 tumors) for CD3[+] T cells in WP, WP-Myc, WB1P and WB1P-Myc tumors. Scale bars = 50 µm. **D** FACS analysis of leukocytes (CD45[+]) (*n* = 4 animals/genotype, *p* = 0.002), T cells (CD3[+]) (*n* = 4 animals/genotype, *p* = 0.04), NK-cells (CD49b[+]) (*n* = 4 animals/genotype, *p* = 0.05), B-cells (CD19[+]) (*n* = 4 animals/genotype, *p* = 0.63), myeloid

cells (CD11b[+]) (*n* = 4, p = 0.002), Monocytes (Ly6C[+]) (*n* = 3 (WB1P), *n* = 4 (WB1P-Myc), *p* = 0.33, Neutrophils (Ly6G[+]) (*n* = 3 (WB1P), *n* = 4 (WB1P-Myc), *p* = 0.001 and Macrophages (F4/80[+], CD206[+]) (*n* = 3 (WB1P), *n* = 4 (WB1P-Myc), *p* = 0.002. WB1P-Myc tumors were compared to WB1P tumors (two tailed Student's t-test as data were normally distributed, mean with SEM plotted). **E** Tumors generated by intraductal lenti-Cre injections in B1P (*n* = 4) and B1P-Myc (*n* = 5) mice are analyzed via immunohistochemistry for CD3 expression and quantified by counting positive cells/area. *P* value was calculated using 2 sided Mann-Whitney test, *p* = 0.01, mean with SD plotted. **F** Heatmap showing the gene sets represented by the comparison of WB1P-Myc versus WB1P tumors and sorted cancer cells from GSEA analysis. Normalized enrichment scores are plotted. Source data are provided as a Source Data file.

---

*Brca1^F/F;Trp53^F/F* (B1P) and *Brca1^F/F;Trp53^F/F;Col1a1^{invCAG-Myc-IRES-Luc/+}* (B1P-Myc*)* mice via intraductal injection of a Cre-encoding lentivirus. This resulted again in profound TIL depletion in the MYC-overexpressing B1P tumors (Fig. 2E).

While WB1P and WB1P-Myc tumors both resemble TNBCs, the latency from tumor induction to outgrowth is greatly reduced upon MYC overexpression[20]. Importantly, WB1P and WB1P-Myc tumors did not show a difference in growth speed once they were palpable (Supplementary Fig. S2F). Dividing Ki67-positive cells are more abundant in WB1P-Myc tumors, but this growth advantage is counterbalanced by a higher apoptosis rate (Supplementary Fig. S2G, H). The higher percentage of apoptotic cells in WB1P-Myc tumors also shows that lower percentages of infiltrating immune cells in these tumors (Fig. 2D) are not simply due to a larger fraction of live tumor cells in WB1P-Myc tumors versus WB1P tumors.

To further exclude that the different tumor latency times in WB1P-Myc mice compared to WB1P mice were playing a role in the differences of immune cell infiltration, we generated *WapCre;Brca1^F/F;Trp53^F/F;Col1a1^{invCAG-Met-IRES-Luc/+}* (WB1P-Met) mice with tumor-specific overexpression of MET instead of MYC. WB1P-Met mice showed a very similar tumor latency in comparison to the WB1P-Myc model (Supplementary Fig. S2I). In contrast to MYC, MET overexpression did not result in immune suppression, as demonstrated by the comparable numbers of TILS in WB1P-Met versus WB1P tumors (Supplementary Fig. S2J). Also clustering based on expression of ISGs resulted in a clear separation between WB1P-Met and WB1P-Myc tumors (Supplementary Fig. S3A). To further examine if the immune cell exclusion in WB1P-Myc tumors was not a generic consequence of tumor-promoting mutations, we tested if loss of an unrelated tumor suppressor, *Pten*, would also lead to decreased lymphocyte infiltration. To this end, intraductal injections were performed in *WapCre;Brca1^F/F; Trp53^F/F;Col1a1^{invCAG-Cas9-IRES-Luc/+}* (WB1P-Cas9) mice with lentiviruses encoding a *Pten*-targeting sgRNA alone (Lenti-sgPten) or in combination with MYC (Lenti-sgPten-Myc)[20]. TILs were observed in tumors from mice injected with Lenti-sgPten, but not in tumors from mice injected with Lenti-sgPten-Myc, confirming that MYC is selectively responsible for immune cell exclusion. The fact that the immune infiltration in WB1P-Cas9 mice is comparable to WB1P mice and the reduction of infiltration upon MYC expression is unaffected by CAS9 further corroborates the notion that the observed phenotype is MYC specific (Supplementary Fig. S3B).

### MYC drives immune cell exclusion in a tumor cell-intrinsic manner

To investigate how MYC is linked to an immune-suppressive phenotype, we performed RNA-seq on two different sources of tumor cells. In addition to bulk tumors containing both tumor cells and infiltrating immune cells, we used FACS-sorted E-cadherin-positive (ECAD[+]) tumor cells from WB1P and WB1P-Myc tumors. Consistent with our analysis mentioned above, GSEA showed significant downregulation of immune response pathways in the bulk tumor samples (Fig. 2F).

Although such transcriptomic changes could be due to the decreased presence of immune cells in bulk tumor samples, these immune pathways were also downregulated in sorted tumor cells, indicating that MYC-associated immune evasion is mediated by a tumor cell-intrinsic mechanism (Fig. 2F). In support of this notion, the enriched pathways showed strong correlation (R = 0.83) between WB1P-Myc bulk tumors and sorted tumor cells (Supplementary Fig. S3C), further underscoring that MYC suppresses IFN signaling in a tumor cell-intrinsic manner.

To corroborate on our preclinical in vivo findings, we used CIBERSORT analysis[34] on gene expression data from TCGA to estimate the fractions of different immune cell types in human breast cancer samples. Compared to cancers with neutral copy numbers of *MYC*, breast cancers with amplified *MYC* contained lower fractions of monocytes, M2 macrophages and CD8[+] T cells, whereas they showed increased fractions of M0/M1 macrophages and regulatory T cells (Supplementary Fig. S3D). A similar pattern was observed within the TNBC subset of breast cancers (Supplementary Fig. S3D). Taken together, our results show that MYC expression drives a dramatic loss of lymphocytic infiltration, as well as other immune cells, in mouse and human breast cancer. Furthermore, we demonstrate a cancer cell-intrinsic role for MYC in suppressing inflammatory pathways.

### MYC overexpression in mammary tumor cells downregulates IFN-stimulated genes

The main downregulated pathways in the MYC-overexpressing WB1P tumors were 'IFN signaling' and 'JAK/STAT signaling' (Fig. 2A, F), which are both important in inflammatory responses[35]. Recent studies showed that loss of BRCA1 leads to accumulation of cytosolic DNA, thereby triggering the cGAS/STING pathway[13,36], (Heijink et al.[14]). We derived organoids from WB1P and WB1P-Myc tumors to investigate whether suppression of IFN signaling in WB1P-Myc tumors is connected to reduced cGAS/STING pathway activation. We first probed RNA-seq profiles from bulk tumors, sorted cells and organoids with a previously reported panel of ISGs induced by cGAS/STING signaling[16]. The expression of these ISGs clearly separated WB1P-Myc from the WB1P tumors and organoids, showing significant downregulation of the ISGs (Fig. 3A). Specifically, we observed down-regulation of various STING pathway-related genes, including *Stat1*, *Stat3*, *Ccl20* and *Irf9* (Fig. 3A)[37,38]. We also observed downregulation of *Cd74* and *Ciita*, two genes important for the function of the adaptive immune response via MHC class II signaling[39]. These genes are directly regulated by STAT1, which has reduced phosphorylation upon MYC expression (Supplementary Fig. S4A, B). Signaling downstream of cGAS/STING is associated with the secretion of different chemokines and cytokines, including CCL5 and CXCL10[16,40]. In line with this notion, by performing a cytokine array and ELISAs on supernatant from WB1P and WB1P-Myc derived organoids as well as qRT-PCRs on the organoids, we found that MYC-expressing organoids show reduced secretion and expression of CXCL10 and CCL5 (Fig. 3B–D, Supplementary Fig. S4C). To confirm the direct role of MYC in the downregulation of ISGs, we transduced WB1P

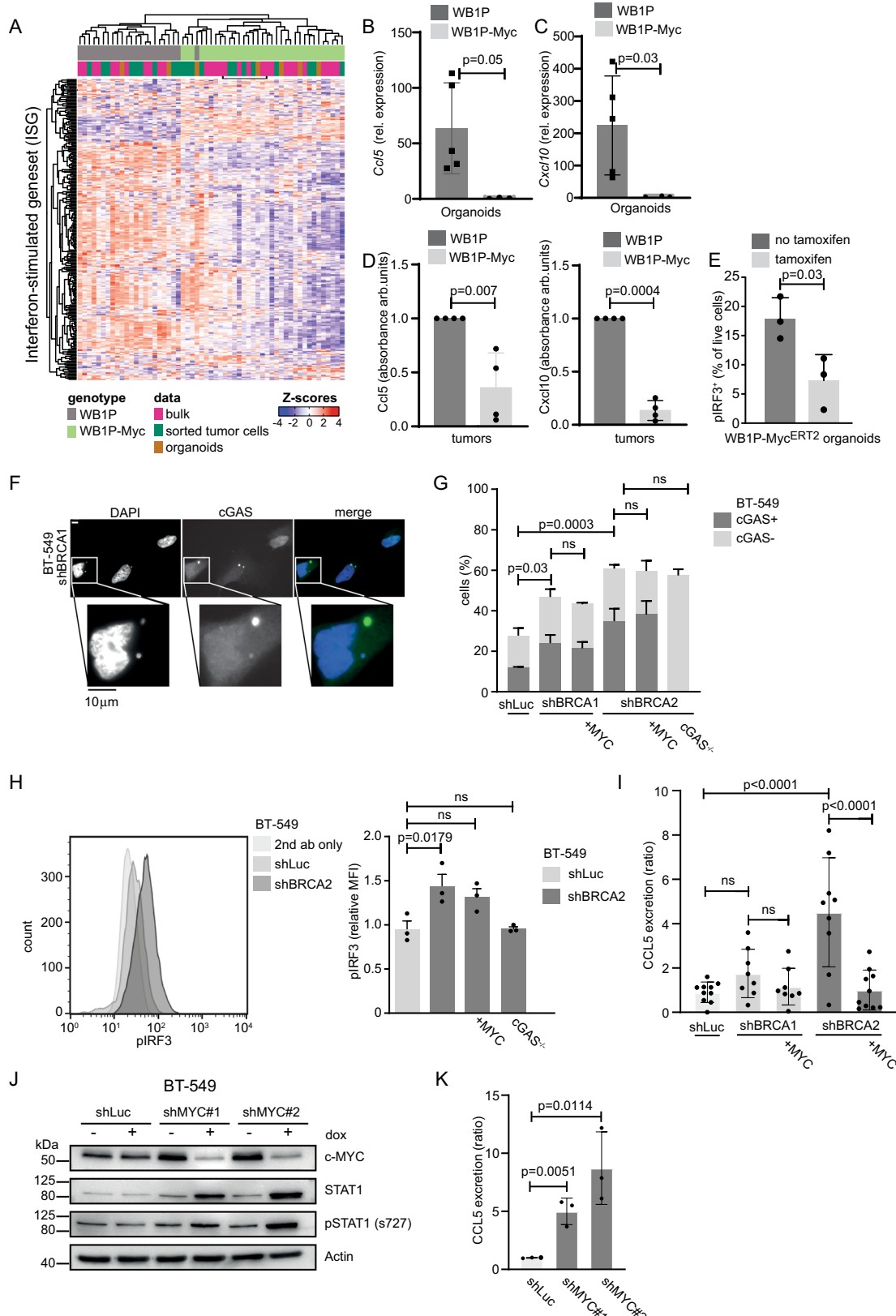

organoids with a lentiviral vector encoding a tamoxifen-inducible MYC^{ERT2} fusion protein. Flow cytometry analysis of these organoids showed that MYC activation upon addition of tamoxifen decreased phosphorylation of Interferon Regulatory Factor 3 (pIRF3) (Fig. 3E, Supplementary Fig. S4D), a key transcriptional regulator of IFN and STING responses. Also phosphorylation of Tank binding kinase

(pTBK1), a central player in the STING signaling pathway was reduced upon MYC activation (Supplementary Fig. S4E). We conclude that MYC overexpression can manipulate IFN signaling by reducing the expression of a broad network of genes in our murine tumor model.

To investigate the effects of MYC overexpression on cGAS/STING signaling in human breast cancer, the TNBC cell line BT-549 was

**Fig. 3 | MYC overexpression in mammary tumor cells down-regulates IFN-stimulated genes. A** Heatmap depicting previously reported interferon-stimulated genes[16] in RNA-seq of bulk tumors, sorted epithelial tumor cells and organoids. **B** qRT-PCR for *Ccl5* in a WB1P ($n = 5$) and a WB1P-Myc ($n = 3$) derived mouse tumor organoid line. Relative gene expression levels normalized to RPM20 (unpaired two-tailed student's t-test, $p < 0.05$, mean with SD plotted). **C** qRT-PCR analysis for *Cxcl10* in a WB1P ($n = 5$) and a WB1P-Myc ($n = 4$) derived mouse tumor organoid line, relative gene expression levels normalized to RPM20 are plotted (unpaired two-tailed student's t-test, $p < 0.05$, mean with SD plotted). **D** ELISA for CCL5 and CXCL10 from supernatant collected 48 h after seeding of equal numbers of dissociated organoids derived from WB1P and WB1P-Myc tumors ($n = 3$, absorbance in arbitrary units (arb. units), normalized to WB1P, unpaired two-tailed t-test, $p < 0.05$, mean with SD plotted). **E** Quantification of pIRF3 by FACS analysis of WB1P-Myc^ERT2 organoids with and without tamoxifen treatment. ($n = 3$, unpaired two-tailed t-test, $p < 0.05$, mean with SD plotted). **F** Representative images of BT-549 cells harboring shBRCA2 and treated with doxycycline (dox) for three days. Cells were stained with anti-cGAS and DAPI. Scale bars =10 μm. **G** Quantification of cGAS-positive micronuclei as described in **F**. ≥ 100 Cells were counted per condition. Error bars indicate

SEM of at least three independent experiments (one-way ANOVA, with Sidak's multiple comparison test). **H** Left panel: BT-549 cells with indicated hairpins, were treated with doxycycline for 5 days. Levels of pIRF3 were analyzed by FACS. Right panel: Quantification of median fluorescence intensities (MFI) were normalized to cells without doxycycline. Error bars indicate SEM of 3 independent experiments (one-way ANOVA, with Sidak's multiple comparison tests). **I** BT-549 cells with indicated hairpins, with overexpression of MYC or deletion of cGAS with or without doxycycline (dox) for 6 days. Secretion of CCL5 was measured with ELISA. Error bars indicate SEM of $n = 8$ (shBRCA1), $n = 9$ (shBRCA2) or $n = 10$ (shLuc, shBRCA2-MYC) independent experiments (one-way ANOVA, with Sidak's multiple comparison tests). **(J)** BT-549 cells with indicated Luc or MYC shRNAs with or without doxycycline (dox) for 3 days. Immunoblotting was performed for c-MYC, STAT1 and pSTAT1. **K** BT-549 cells with MYC shRNAs with or without doxycycline (dox) for 5 days. Secretion of CCL5 was measured with ELISA. Concentrations were normalized to untreated conditions. Error bars indicate SEM of 3 independent experiments (unpaired two-tailed t-test). Source data are provided as a Source Data file.

transduced with doxycycline-inducible short hairpin RNAs (shRNA) targeting BRCA1 or BRCA2, with or without constitutive over-expression of MYC (Supplementary Fig. S4F), whereas cGAS^−/− cells served as controls. Of note, shRNA-mediated depletion of BRCA1 and BRCA2 resulted in decreased cell proliferation and ultimately cell death (Supplementary Fig. S4G–I), which was not rescued by MYC overexpression (Supplementary Fig. S4G–I). In line with previous reports[16], (Heijink et al.[14]), BRCA1 and BRCA2 depletion led to increased amounts of cGAS-positive micronuclei (Fig. 3F, G). This increase was not suppressed by MYC overexpression (Fig. 3G), suggesting that the role of MYC in suppressing IFN signaling acts downstream of the generation of cytoplasmic DNA. In line with this, whereas pIRF3 levels were increased upon BRCA2 depletion, pIRF3 levels were only marginally affected by MYC overexpression, supporting a more downstream mode of action than micronuclei formation and the resulting pIRF3 activation in BT-549 cells (Fig. 3H, Supplementary Fig. S5A), suggesting differences between the human TNBC cells and mouse mammary tumor organoids.

In line with our observations in WB1P and WB1P-Myc mammary tumors, expression of CCL5 and pSTAT1 was induced upon BRCA1/2 depletion, and was suppressed in MYC-overexpressing BT-549 cells (Fig. 3I, Supplementary Fig. S4J). Of note, BRCA2 depletion had a stronger effect in this model than BRCA1 depletion (Fig. 3I, Supplementary Fig. S4J). Conversely, CCL5 secretion as well as STAT1 expression and phosphorylation were increased upon depletion of MYC in BT549 cells using doxycycline-inducible shRNAs, supporting the notion of MYC having immune suppressive capacity (Fig. 3J, K, Supplementary Fig. S5B).

## MYC status of breast cancer cells regulates lymphocyte trafficking and activation in vitro and in vivo
To further assess the role of MYC in the inhibition of inflammatory signaling and thereby suppressing immune responses, we turned to human and mouse in vitro systems. Using trans-well assays, we measured the migration of isolated human CD8^+ T cells towards BT-549 cells upon BRCA1 or BRCA2 depletion (Fig. 4A). Upon BRCA1/2-depletion for 24 h, we observed increased numbers of CD8^+ T cells that migrated towards the BT-549 tumor cells, which was decreased upon MYC overexpression (Fig. 4B). Of note, depletion of BRCA2 was more efficient than that of BRCA1 (Supplementary Fig. S4J). In parallel, we harvested conditioned medium to probe whether secreted factors upon BRCA1/2 depletion conferred an ability to stimulate CD8^+ T cell proliferation and activation (Supplementary Fig. S5C). Overexpression of MYC only induced a weak and non-significant inhibitory effect on activation and proliferation of T cells in this experimental setting (Supplementary Fig. S5C). These results suggest that factors secreted by BRCA1- or BRCA2-depleted breast cancer cells promote the

migration rather than activation of T cells via activation of STING signaling. In contrast, MYC overexpression suppresses the migration of CD8^+ T cells in a tumor cell-intrinsic manner.

To confirm that MYC overexpression has similar effects in mouse mammary tumor cells, we performed live-cell imaging in co-cultures of WB1P and WB1P-Myc tumor-derived organoids and syngeneic mouse splenocytes as well as isolated CD8^+ T cells. 7-day time-lapse quantification of organoid size demonstrated clear growth inhibition of WB1P organoids by immune cells, whereas immune cells affected WB1P-Myc organoids only to a lower degree (Fig. 4C). Also, proliferation analysis in these co-cultures demonstrated that WB1P-Myc organoids suppressed IL2-induced T cell proliferation (Fig. 4D). Since MYC expression leads to reduction of chemokine secretion (Fig. 3D, Supplementary Fig. S4C), we hypothesized that we could restore T cell killing by adding CCL5 and CXCL10 to the culture medium. Indeed, addition of these chemokines resulted in increased killing mediated by isolated CD8^+ T cells in the WB1P-Myc – CD8^+ T cell co-cultures (Fig. 4E).

To investigate whether MYC directly controls immune cell tumor infiltration in vivo, we assessed the effects of MYC-activation or -inactivation in established BRCA1-deficient mouse mammary tumors in situ. To this end, we generated *WapCre;Brca1^F/F;TrpS3^F/F;Col1a1^invCAG-MycERT2-IRES-Luc/+* (WB1P-Myc^ERT2) GEMMs and investigated tumor growth and immune infiltration at different time points after MYC induction via tamoxifen administration. Upon feeding tamoxifen at seven weeks of age until the time of sacrifice, we again observed that MYC induction reduced immune cell infiltration and shortened tumor latency compared to WB1P tumors, similar to findings with the WB1P-Myc model (Fig. 4F, G and Supplementary Fig. S6A). Also, we observed higher numbers of tumors, as expected for MYC-driven tumorigenesis (Fig. 4H). Whereas tumor growth rates were not significantly altered upon tamoxifen-induced Myc^ERT2 translocation in already established tumors, MYC activation resulted in depletion of immune infiltrates (Fig. 4I). Specifically, a time-dependent reduction in CD3^+ T cells was observed after tamoxifen administration until they resembled the low levels that were observed in tumors in the WB1P-Myc mice (Fig. 4I, Supplementary Fig. S6A). Importantly, CD3^+ T cell numbers were not affected by the size of the tumor (Supplementary Fig. S6B). MYC-induced reduction of TILs was also observed in B1P-Myc^ERT2 tumors induced by intraductal injection of lentiviral Myc^ERT2-P2A-Cre as well as in orthotopically transplanted WB1P tumor-derived organoids that were transduced with lentiviral Myc^ERT2, where we also observed a correlation of tumor growth inhibition with immune infiltration upon MYC withdrawal (Supplementary Fig. S6C–E). Combined, these findings confirmed our previous results, where we saw that MYC expression directly hinders immune infiltration in BRCA1-mutant tumors and underscore that the decreased immunogenicity does not result from different tumor latencies or sizes.

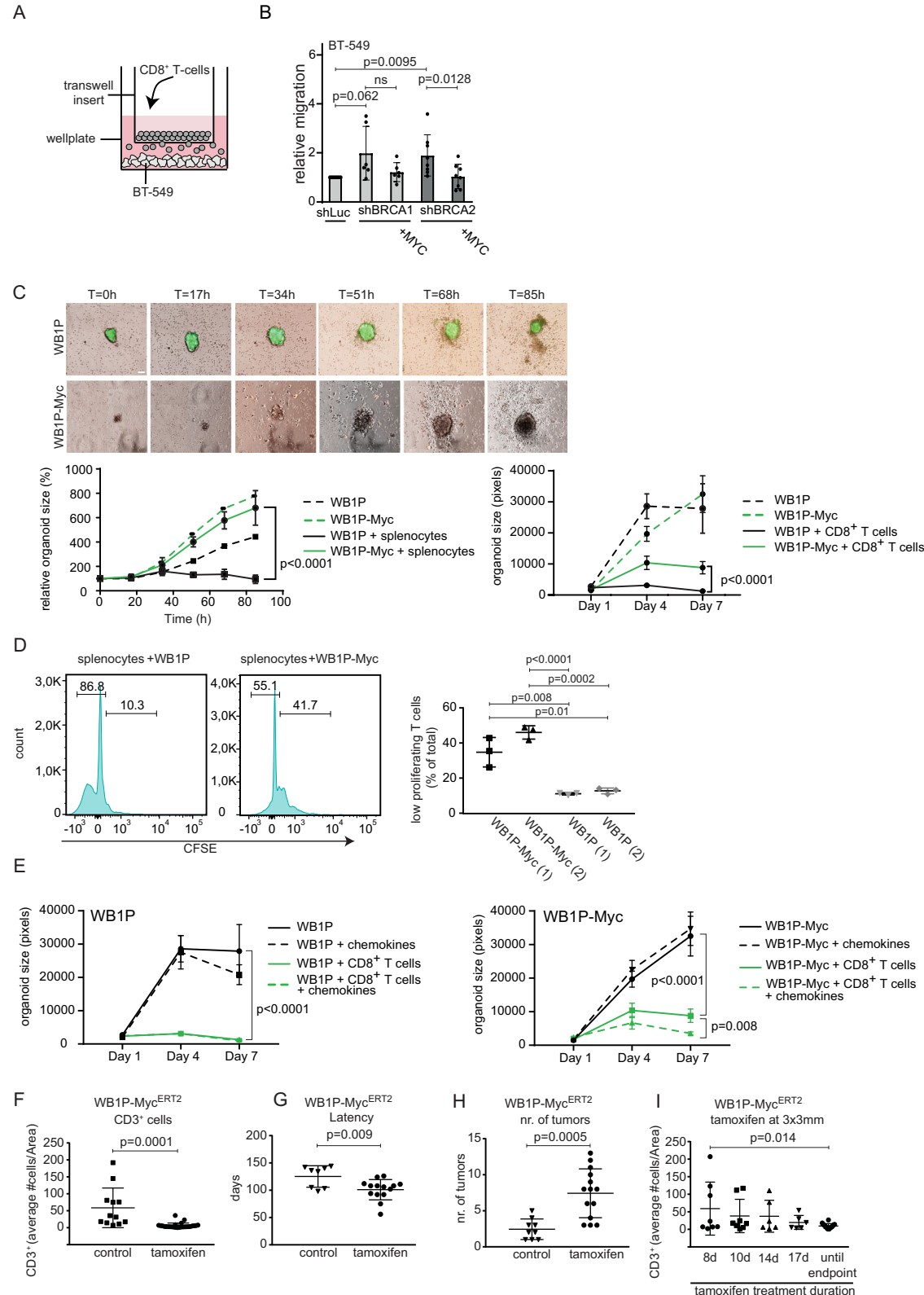

## MYC controls expression of multiple IFN signaling components in tumors and organoids

To test if MYC directly regulates genes involved in IFN signaling, we performed chromatin immuno-precipitation of MYC, followed by massive parallel sequencing (ChIP-seq) on WB1P and WB1P-Myc organoids as well as tumors. We found 1257 shared peaks between the tumor and organoid ChIPs (Supplementary Fig. S7A), of which the majority was found in promoter regions (Supplementary Fig. S7B).

MYC binding was significantly enriched in the promoter regions of those genes that were previously found to be up-regulated in the RNAseq of WB1P-Myc bulk tumors, sorted tumor cells and organoids (Fig. 5B and supplementary Data 1). We retrieved the MYC motif in the majority of peaks (Supplementary Fig. S7E).

Next, we intersected the MYC-bound genes identified by ChIP-seq with the genes downregulated upon MYC overexpression in the mouse mammary tumors, organoids and sorted tumor cells

**Fig. 4 | MYC directly regulates lymphocyte trafficking and activation in vitro and in vivo. A** Schematic overview of the transwell assay. BT-549 cells were cultured for 5 days with doxycycline to induce expression of indicated shRNAs. Human CD8$^+$ T cells were added for 24 h. T cells migrated towards the lower compartment were counted. **B** Transwell assays were performed as described in A. Migrated T cells were counted after 24 h. Data was normalized to shLUC values. Data of $n = 6$ (shBRCA) or $n = 8$ (shLuc/shBRCA2) independent experiments are shown (one-way ANOVA, with Sidak's multiple comparison tests). **C** Live imaging of WB1P (green) and WB1P-Myc organoids (red) with splenocytes. Time after seeding in hours is indicated. Quantification of tumor organoid sizes is depicted in the lower left panel (scale = 100 μm) ($n = 3$, two-way ANOVA, $p < 0.0001$). The lower right panel shows in vitro co-culture of WB1P and WB1P-Myc organoids with CD8$^+$ T cells. Organoid sizes were measured at day 1, 4 and 7 ($n = 3$, representative experiment shown, two-way ANOVA, $p < 0.0001$, mean with SD plotted). **D** FACS analysis of WB1P and WB1P-Myc organoid-splenocyte co-cultures stained for CSFE. Halving of fluorescence-intensity marks one cell division. Fluorescence-intensity per number of cells is plotted. Percentage of low proliferating T cells of 2 different organoid lines/condition is shown in the right panel. (unpaired two tailed t-tests, mean with SD plotted). **E** Co-cultures of WB1P (left) and WB1P-Myc (right) organoids with and without CD8$^+$ T cells and chemokines (CXCL10 and CCL5). Organoid sizes were measured at day 1, 4 and 7 (two-way ANOVA, for WB1P+/− CD8$^+$-T cells $p < 0.0001$; with versus without chemokines $p = 0.794$ (ns); for WB1P-MYC+/− CD8-T cells $p < 0.0001$; with versus without chemokines $p = 0.0081$, $n = 3$ organoid wells/genotype and condition, repeated 3 times independently, representative experiment shown, mean with SD plotted). **F** Counts/Area of CD3$^+$ T cells of mammary tumors in the Myc$^{ERT2}$ GEMM with ($n = 34$) and without tamoxifen chow ($n = 12$) until endpoint (tumor 15 mm x 15 mm) (unpaired two-tailed t-test, $p = 0.0001$, mean with SD plotted). **G** Tumor latency in WB1P-Myc$^{ERT2}$ mice with ($n = 14$) and without tamoxifen chow ($n = 8$), (unpaired two-tailed student's t-test, $p = 0.009$, mean with SD plotted). **H** Tumor burden in mice with ($n = 14$) and without tamoxifen ($n = 9$), (unpaired two-tailed student's t-test, $p = 0.0005$, mean with SD plotted). **I** Counts/Area of CD3$^+$ T cells of mammary tumors in the Myc$^{ERT2}$ GEMM upon tamoxifen administration at $3 \times 3$ mm tumor-volume (unpaired two-tailed student's t-test, Tamoxifen for 8 days ($n = 8$) versus until endpoint ($n = 16$), (unpaired two-tailed student's t-test, $p = 0.014$, mean with SD plotted). Source data are provided as a Source Data file.

(Fig. 5A, Supplementary Fig. S7C, D). GSEA revealed that genes involved in IFN signaling and inflammation, including 'JAK STAT3 signaling' (Fig. 5C) were enriched among the 129 genes that were bound by MYC and downregulated upon MYC overexpression. MYC ChIP-seq peaks that occurred in tumors and/or organoids overlapped with 59 IFN signaling pathway-associated genes that were repressed by MYC in the tumors, sorted tumor epithelial cells and/or organoids (Supplementary Fig. S7D). We next constructed a co-functionality network[41] using all 129 MYC-repressed genes (Fig. 5A), which revealed a network of immune- and IFN-related genes among the MYC target genes, again confirming the role of MYC in suppressing inflammatory signaling (Supplementary Fig. S7F). Conversely, a co-functionality network for MYC-upregulated genes did not show any immunity signatures (Supplementary Fig. S7E). To confirm that the observed effects were indeed MYC-mediated repression of immune modulatory genes, we made use of the MYC mutant V394D, which abrogates MYC binding to its co-repressor MIZ1[42]. Tumors induced in B1P animals via intraductal injections of lentiviruses encoding Cre together with MYC-V394D have a longer latency than tumors induced with lentiviruses encoding Cre together with wild-type MYC (Fig. 5D). Upon overexpression of MYC-V394D, tumors also show increased immune cell invasion when compared to tumors expressing wild-type MYC (Fig. 5D). Next, we performed ChIP-seq on WB1P-Myc tumors with MIZ1, the co-repressor needed for MYC-mediated repression[24]. We found a high overlap of MIZ1 with MYC binding sites of 7,654 peaks (Supplementary Fig. S8A, B), while overlapping peaks were predominantly located in promoter regions (Supplementary Fig. S8C, D). GSEA after overlaying the MIZ1-MYC shared peaks with the genes downregulated in the WB1P-Myc RNA-seq datasets as well as construction of another co-functionality network yielded innate immunity signatures, confirming the direct repression of immunity genes by MYC (Fig. 5E, F, Supplementary Fig. S8E, Supplementary Data 2). Combined, our results demonstrate that MYC directly controls immune infiltration into tumors via downregulation of a myriad of inflammatory pathway components.

## MYC suppresses anticancer efficacy of pharmacological STING activation in WB1P tumors

To further investigate how MYC influences inflammatory signaling, we used the small molecule STING agonist vadimezan (DMXAA) to activate IFN signaling in mice with established WB1P and WB1P-Myc tumors. WB1P and WB1P-Myc organoids were transplanted into mammary fat pads of immunocompetent syngeneic mice. Additionally, lentivirus encoding Cre was injected intraductally into WB1P-Myc mice. When tumors reached a volume of 100 mm³, vadimezan was administered every 14 days for 6 weeks. Vadimezan treatment induced

transient regression and stabilization of WB1P tumors until treatment stop, after which tumor growth resumed (Fig. 6A, left panel). WB1P-Myc tumors were more resistant to the effects of vadimezan-mediated STING activation than WB1P tumors, as tumor regression was not observed in this setting (Fig. 6A, right panel). Of note, vademizan-treated WB1P-Myc tumors still showed a clear growth delay in comparison to vehicle-treated tumors (Fig. 6B), as well as influx of immune cells into tumors upon treatment (Fig. 6C, D). Similarly, STING-agonist treatment of co-cultures of tumor organoids with splenocytes strongly blocked growth of WB1P organoids as measured by MTT assay, whereas WB1P-Myc organoids were largely unaffected, confirming our in vivo experiments. Of note, WB1P organoids co-cultured with splenocytes but without STING activation do not show a growth reduction in the MTT assay, presumably because the metabolic activity of the splenocytes in the culture is also included in the measurement (Fig. 6C). Together, these findings show the relevance of IFN signaling in suppressing tumor growth in TNBCs and underscore the critical role of MYC in suppressing the IFN response via direct inhibition of immunomodulatory factors downstream and in parallel of cGAS/STING signaling. Our findings also highlight the difficulty of countering the broad immuno-inhibitory effects of MYC by therapeutic targeting of a single downstream factor such as STING (Fig. 6D).

## Discussion

It remains enigmatic why TNBCs and *BRCA*-mutated breast cancers rarely respond to immunotherapy (Voorwerk et al.[30]), despite these cancers being viewed as immunogenic due to the increased amount of neo-antigens induced by their genomic instability[43–46]. In this study, we show that a substantial fraction of TNBCs may evade immune clearance via MYC overexpression. We demonstrate that MYC suppresses immune cell infiltration into tumors by repressing IFN/STING signaling in a tumor cell-intrinsic manner. Our findings provide one mechanism by which TNBCs and *BRCA*-mutated breast cancers may evade clearance by the immune system.

Suppression of immune responses against tumor cells is of great importance for cancer development and progression. Consequently, therapeutic boosting of the adaptive immune system via inhibition of immune checkpoint components PD1/PD-L1 and CTLA4 has been successfully used to activate T cell responses against tumors[47]. Surprisingly, the anti-PD-1 inhibitor pembrolizumab was effective in only 20% of TNBC patients as monotherapy (Voorwerk et al.[30]), despite the fact that TNBCs are predicted to be immunogenic due to high levels of genomic instability[48,49]. Also, no association was found between response to pembrolizumab and *BRCA1/2* mutations, and BRCA1-like genomic copy number profiles were even negatively associated with response (Voorwerk et al.[30]). This is especially puzzling since loss of

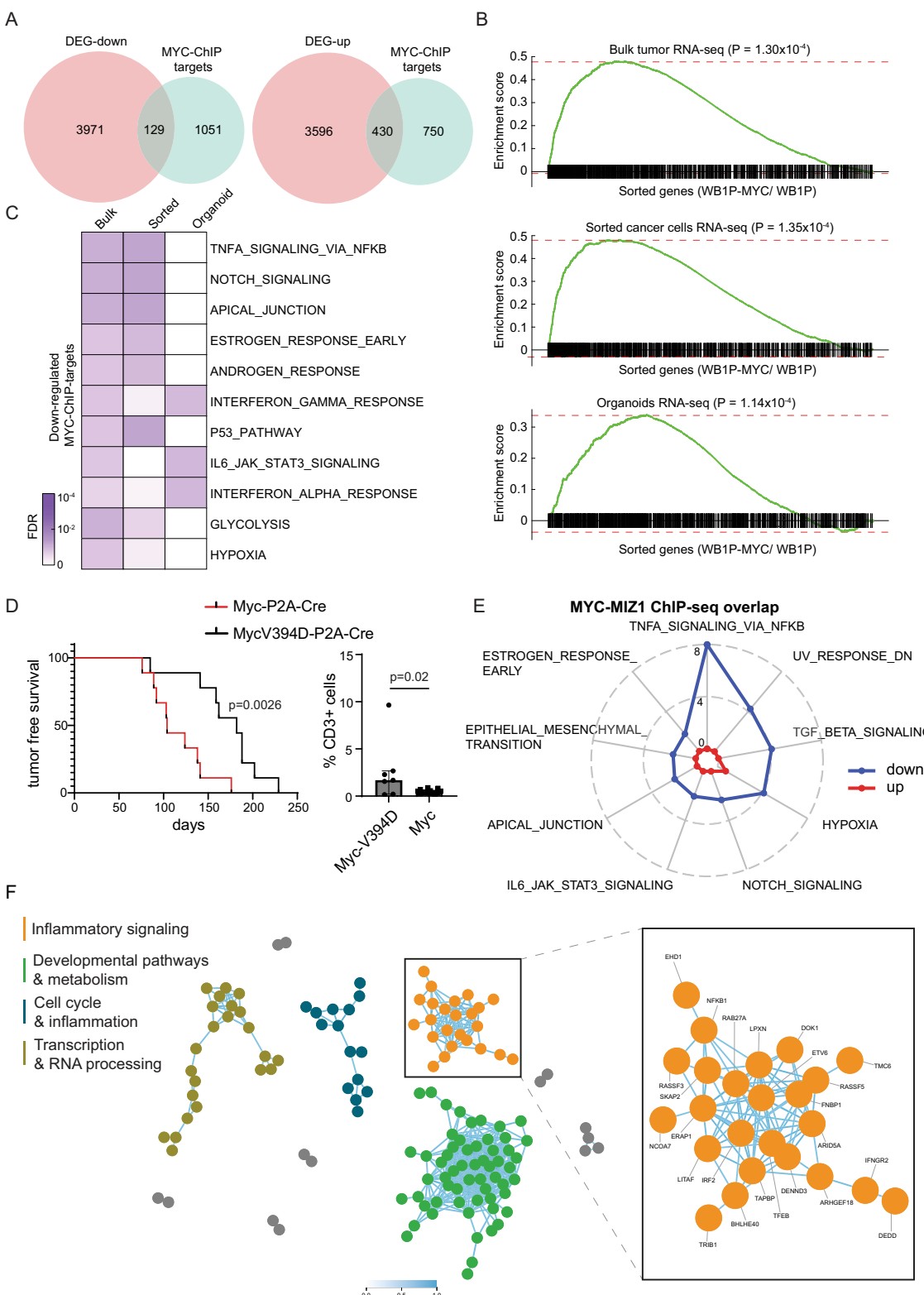

BRCA1/2 function results in activation of STING signaling and subsequent attraction of immune cells in patients[13,14,50]. Of note, whereas single-agent ICI did not show efficacy in TNBC, neo-adjuvant paclitaxel-based treatment with ICI did show some efficacy in patients with TNBC[51]. Possibly, baseline inflammatory signaling in TNBC cells is suppressed and precludes local activation of T cells. Treatment with chemotherapeutic agents that trigger inflammatory signaling, including paclitaxel, may trigger sufficient interferon signaling in TNBC tumor cells to allow ICI-mediated immune responses[29,52]. Our results

suggest that the efficacy of immune checkpoint inhibitor (ICI) therapy in TNBC and *BRCA*-mutated breast cancer may be restrained by MYC-induced suppression of local immune cells. This may be the case in a substantial fraction of TNBCs, as our analysis of oncogene amplification in a cohort of 142 human *BRCA*-mutant TNBCs from the TCGA dataset revealed *MYC* amplifications in the majority of cases, far more often than any other oncogene. GSEA of RNA-seq data from the same cohort showed association of *MYC* amplification with reduction in immune signatures, specifically IFN and inflammatory signaling.

**Fig. 5 | MYC controls expression of multiple IFN signaling components in tumors and organoids. A** Overlap between MYC target genes from ChIP-seq and differentially expressed genes (DEGs) comparing WB1P-Myc with WB1P. For this comparison, MYC targets were obtained from the common MYC-binding loci from tumor and organoid ChIP-seq data, and DEGs were obtained from the union of the genes showing differential expression between WB1P-Myc versus WB1P in bulk tumor, sorted tumor cells, and organoid RNA-seq data (see Methods). **B** GSEA analysis of MYC targets in each bulk tumor, sorted cancer cells, and organoid RNA-seq data comparing WB1P-Myc to WB1P. The genes closest to common MYC-binding loci between tumor and organoid ChIP-seq defined MYC targets. **C** Gene sets significantly over-represented by the down-regulated MYC targets from Fisher's exact test (FDR < 0.1). The down-regulated MYC targets were defined by the genes in Fig. 5A (129 down-regulated MYC targets). **D** Left: Kaplan-Meyer curve of mammary tumor latency analysis of B1P mice injected with lenti-viruses encoding Cre-P2A-MYC-V394D and Cre-P2A-MYC (log rank (Mantel-Cox) test, $n = 9$ mice/

genotype, p = 0.0026). Right: Immunohistochemistry was performed to analyze CD3[+] T cell numbers using quPath software (unpaired two-tailed Kolmogorov-Smirnov test to check for differences in variability between groups, $n = 7$ (Myc-V394D) and 14 (wild-type Myc), $p = 0.02$, median with interquartile range plotted). **E** Spider plot of gene sets significantly over-represented by the downregulated MYC targets within the overlapping peaks of MYC and MIZ1 ChIP-seq in the promoter regions. The numbers on the lines of the plot indicate $-\log_{10}(FDR)$. **F** Constructed co-functionality network of genes downregulated by MYC ($n = 250$) retrieved from both MYC-ChIP-seq and MIZ1 ChIP-seq of WB1P-Myc tumors as well as RNA-seq data of WB1P and WB1P-Myc tumors and organoids. Right: one of four identified clusters with strong shared predicted co-functionality ($r > 0.5$) shows enrichment for inflammatory pathways (e.g. innate immune system, Interferon signaling and cytokine signaling in immune system). Detailed lists of enriched pathways can be found in Supplementary Fig. S8E. Source data are provided as a Source Data file.

Complementary GSEA of a cohort of samples from the TONIC trial (Voorwerk et al.[30]) confirmed the correlation of high MYC expression with reduced immune signatures, in line with previous findings, where *MYC* amplifications were correlated with 'immune deserts' in TNBCs (Xiao et al.[29]).

Previous studies have shown that the effects of MYC on the tumor immune microenvironment differ between cancer types. For example, MYC activation induces CCL9-mediated attraction of macrophages in both lung adenocarcinoma (LUAD) and pancreatic ductal adenocarcinoma (PDAC). Still, it has opposite effects on B cells in these cancer types, driving their expulsion in LUAD and influx in PDAC[25,27]. Another study reported that MYC activation in PDAC represses IFN signaling and invasion of NK and B cells via transcriptional repression of IRF5/7 and STAT1/2[28]. Together, these studies suggest that MYC has tissue-specific effects on distinct immune cell populations via defined signaling pathways. Our study shows that in triple-negative and *BRCA1/2*-mutated breast cancer, MYC utilizes yet another strategy to promote immune evasion, namely by expulsion of virtually all tumor-infiltrating immune cell populations.

Our data also show that MYC does not act via a single downstream target, but instead functions as a master regulator, inhibiting numerous effectors of multiple immune-signaling cascades. Specifically, we find that MYC, in conjunction with the transcriptional repressor MIZ1, represses transactivation of a range of inflammatory genes. Indeed, expression of a MYC mutant that is unable to bind MIZ1 showed strongly reduced immune repulsive effects.

To enhance the efficacy of ICI therapy, an obvious goal is turning immunologically "cold" tumors into "hot" tumors[53,54]. A potential strategy to bypass the immune-suppressive effects of MYC over-expression in triple-negative and *BRCA*-mutated breast cancer would be to activate IFN signaling via STING agonists in tumor cells and their environment[55,56]. However, while boosting STING signaling inhibits growth of TNBCs that do not overexpress MYC, its impact on MYC-overexpressing TNBCs is rather moderate, supporting our notion that MYC targets effectors downstream of STING and effectors of other immune pathways. Effective inhibition of the immune-suppressive effects of MYC overexpression in TNBC would therefore require activating innate immunity at a more downstream level via administering interferons or direct targeting of MYC activity, which has proven challenging thus far.

It has recently been reported that not only MYC, but also other oncogenes such as KRAS may have the ability to suppress immune responses against tumors by inhibiting IFN signaling[57]. In line with this, we find that amplifying mutations in several oncogenes in human TNBC leads to markedly decreased expression of immune signature genes, including genes related to IFN signaling.

Taken together, our findings demonstrate a role for MYC in counteracting immune-cell invasion in TNBC via direct inhibition of IFN signaling responses. MYC-induced expulsion of TILs could explain

the ineffectiveness of ICI therapy in a large fraction of triple-negative and *BRCA*-mutated breast cancers that are potentially immunogenic due to their genomic instability. Insight into how IFN signaling is silenced should be incorporated into designing combination therapies to activate IFN signaling and ultimately improve the response rates of MYC-amplified TNBCs to ICIs.

## Methods
### Mice and in vivo procedures
*WapCre;Trp53^{F/F}* (WP), *WapCre;Brca1^{F/F};Trp53^{F/F}* (WB1P), *WapCre;Trp53^{F/F}; Col1a1^{invCAG-Myc-IRES-Luc/+}* (WP-Myc), *WapCre;Brca1^{F/F};Trp53^{F/F};Col1a1^{invCAG-Myc-IRES-Luc/+}* (WB1P-Myc), *WapCre;Brca1^{F/F};Trp53^{F/F};Col1a1^{invCAG-Met-IRES-Luc/+}* (WB1P-Met), and *WapCre;Brca1^{F/F};Trp53^{F/F};Col1a1^{invCAG-Cas9-IRES-Luc/+}* (WB1P-Cas9) mice were generated as described previously[20]. *WapCre;Brca1^{F/F}; Trp53^{F/F};Col1a1^{invCAG-MycERT2-IRES-Luc/+}* (WB1P-Myc^{ERT2}) were generated as described in[20] for WB1P-Myc, the ERT2 was added in frame to the murine *Myc* sequence. Intraductal injections were performed as described[20]. In brief, lentiviral particles were injected intraductally into the mammary glands via the nipple of the mouse. After injection, mice were monitored for mammary tumors twice per week and sacrificed upon reaching humane end-points or tumor size of 1500 mm³. Organoid transplantations into the fat pad of the 4th mammary gland of syngeneic mice were performed as described[20]. For activation of Myc^{ERT2}, Tamoxifen 400-citrate pellets were used as staple chow (Envigo, TD55125). Vadimezan was dissolved in 2.5% bicarbonate at 1 mg/ml and administered at 25 mg/kg every 14 days for 6 weeks when tumors reached 100 mm³. Animals were stratified into the different treatment groups and the treatments were performed by animal technicians blinded regarding the hypothesis of the treatment outcome. All animal experiments were approved by the Animal Ethics Committee of the Netherlands Cancer Institute (Amsterdam, the Netherlands) and performed in accordance with the Dutch Act on Animal Experimentation under CCD licenses AVD301002016407 and AVD3010020172464.

### Human cell lines
Human breast cancer cell lines MDA-MB-231, HCC1806, BT-549 and HCC38 were obtained from ATCC (CRM-HTB-26, CRL-2335, HTB-122, CRL-2314). Breast cancer cell lines were cultured in Roswell Park Memorial Institute (RPMI) medium supplemented with 10% fetal calf serum and penicillin/streptomycin (100 units per mL). Human cell lines were cultured at 37 °C in a humidified incubator with 5% $CO_2$.

### Viral vectors and transduction
For Lenti-Cre-transduction, pBOB-CAG-iCRE-SD (Addgene, plasmid #12336) was used. Lenti-MycP2ACre and Lenti-Myc^{ERT2} P2ACre were cloned as follows: GFP-T2A-puro was removed by AgeI and SalI digest from the SIN.LV.SF-GFP-T2A-puro[58] and P2ACre was inserted as

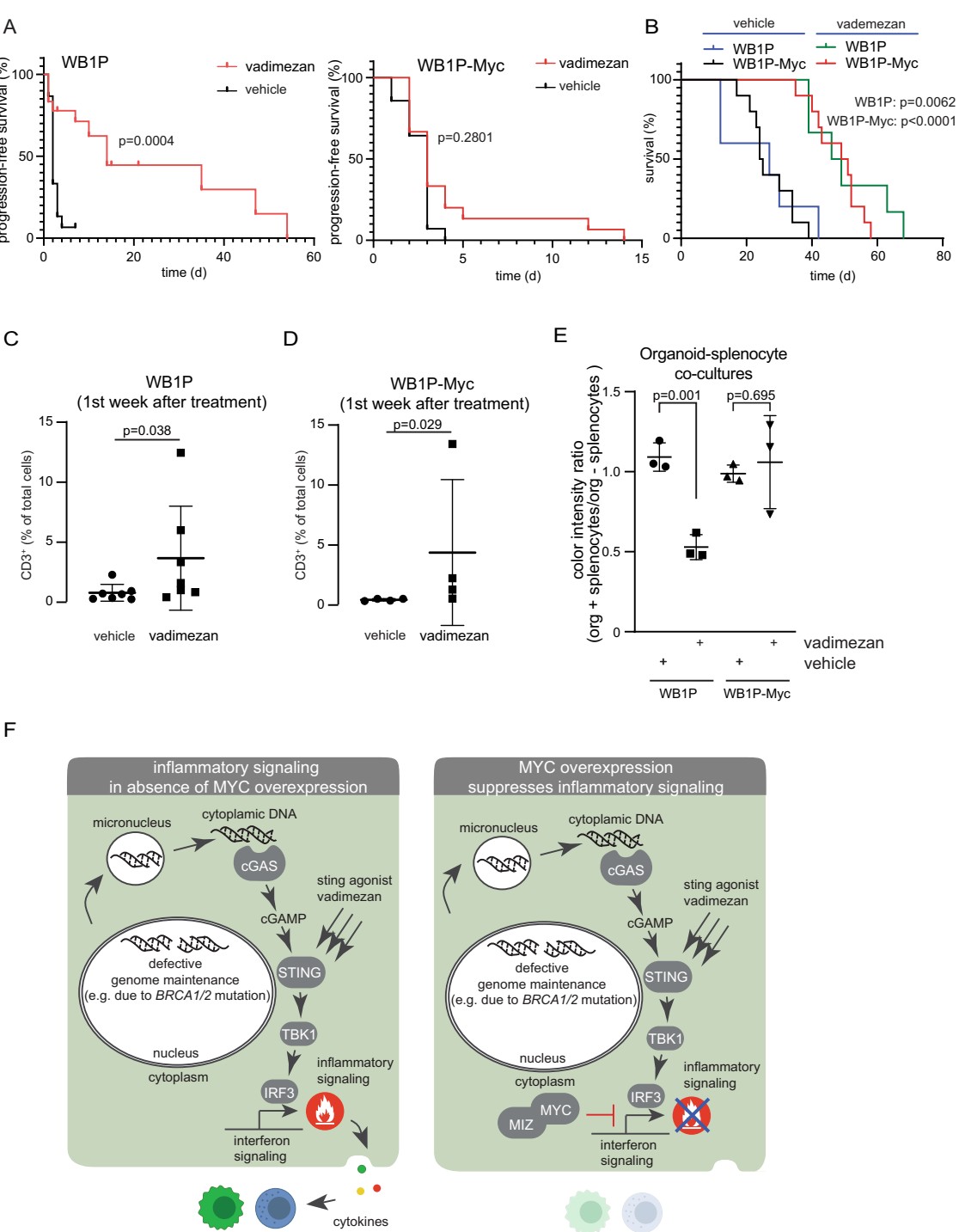

**Fig. 6 | MYC suppresses anti-cancer efficacy of pharmacological STING activation in WB1P tumors. A** Kaplan-Meyer-curves of mice carrying WB1P tumors (left panel) or mice carrying WB1P-Myc tumors (right panel) and treated every 14 days for 6 weeks with vehicle or 25 mg/kg vadimezan i.p when tumors reached 100 mm³. Shown is the time of progression-free survival after treatment ($n = 10$, Log-rank (Mantel-Cox) test). **B** Kaplan-Meyer-curves of mice carrying WB1P tumors or WB1P-Myc tumors, treated as for **A**. Shown is the time of overall survival after treatment ($n = 10$, Log-rank (Mantel-Cox) test, $p = 0.0062$ for WB1P and $p < 0.0001$ for WB1P-Myc). **C** Analysis of CD3+ cell counts in vehicle and vadimezan treated WB1P tumors in the first week after treatment. Plotted are percentages of CD3+ cells/total cells as analyzed by quPath ($n = 7$, unpaired two tailed Mann-Whitney test, $p < 0.05$, mean with SD plotted). **D** Analysis of CD3+ cell counts in vehicle and vadimezan treated WB1P-Myc tumors in the first week after treatment. Plotted are percentages of CD3+ cells/total cells as analyzed by quPath. ($n = 4$, unpaired two tailed Mann-

Whitney test, $p < 0.05$, mean with SD plotted). **E** Organoid-splenocyte co-cultures were checked for viability with an MTT-assay with and without vadimezan treatment (10ug/ml, 7 days culture, $n = 3$). Organoid viability as measured by color intensity from the MTT assay was used as read-out. Plotted is the ratio of color intensity of organoids cultured with splenocytes versus organoids without them, demonstrating that MYC expression protects organoids from splenocyte attack during vadimezan treatments (two tailed unpaired student's t-test, $p < 0.01$, $n = 3$ independent replicates, average of 5 wells/condition/replicate is plotted as mean with SD). **F** Model illustrating how MYC directly suppresses STING/IFN signaling in tumor cells with genomic instability. While genomic instability enhances STING/IFN signaling, no immune response is mounted due to MYC-mediated repression of the downstream effectors, resulting in lack of immune cell recruitment. Source data are provided as a Source Data file.

AgeI-SalI fragment into the SIN.LV.SF-GFP-T2A-puro backbone. The murine *Myc* cDNA was isolated with BamHI-AgeI overhangs using standard PCR from cDNA Clone 8861953 (Source BioScience) and inserted into the SIN.LV.SF-P2ACre vector (all vectors described also in[20]. The MYC-V394D-P2Acre vector was generated by site-directed mutagenesis from the SIN.LV.SF-P2Acre vector using QuickChange lightning kit from Agilent. Lenti-Myc[ERT2] P2A-puro was cloned by inserting the Myc[ERT2] cassette via BamHI and Age1 into the SIN.LV.SF-GFP-T2A-puro backbone. The Lenti-sgPten, Lenti-sgNT, Lenti-sgPten-Myc and Lenti-sgNT-Myc vectors were generated by inserting the *Myc* cDNA with XbaI-XhoI overhangs into the pGIN lentiviral vector for sgRNA overexpression[20]. The non-targeting sgRNA (TGATTGGG GGTCGTTCGCCA) and sgRNA targeting mouse *Pten* exon 7 (CCTCAGC CATTGCCTGTGTG) were cloned as described[59]. Sanger sequencing was used for validation of all vectors. Co-transfection of four plasmids was used to produce concentrated VSV-G pseudotyped lentivirus in 293 T cells[60]. The qPCR lentivirus titration kit from Abm (LV900) was used to determine titers.

To generate dox-inducible knockdown cell lines, BT-549 and HCC38 cell lines were infected with Tet-pLKO-puro harboring short hairpin RNAs (shRNAs). Tet-pLKO-puro was a gift from Dmitri Wiederschain (Addgene plasmid #21915). Hairpin targeting sequences that were used are: BRCA1 (5′-GAG-TAT-GCA-AAC-AGC-TAT-AAT-3′), BRCA2 (5′-AAC-AAC-AAT-TAC-GAA-CCAAAC-TT-3′), MYC#1 (5′-CCC-AAG-GTA-GTT-ATC-CTT-AAA-3′), MYC#2 (5′-CAG-TTG-AAA-CAC-AAA-CTT-GAA-3′) and luciferase ('shLUC', 5′-AAG-AGC-TGT-TTC-TGA-GGA-GCC-3′).

To generate MYC overexpressing cell lines, BT-549 and HCC38 cell lines were infected with retrovirus containing pWZL-Blast-myc. pWZL Blast myc was a gift from William Hahn (Addgene plasmid #10674). Lentiviral and retroviral particles were produced as described previously (Heijink et al.[14]). In brief, 293 T packaging cells were transfected with 10 μg DNA in combination with the packaging plasmids VSV-G and ΔYPR or Gag-Pol and VSV-G complemented with pAdvantage using a standard calcium phosphate protocol. Virus-containing supernatants were harvested and filtered through a 0.45 μM syringe filter with 4 μg per mL polybrene. Supernatants were used to infect target cells in two or three consecutive 24-hour periods. Infected cells were selected in medium containing puromycin (2 μg per mL) or Blasticidin (1 μg per mL) for at least 48 h. Monoclonal cell lines were grown after single-cell sorting. Knock-down or overexpression was confirmed by immunoblotting. For dox-inducible expression of MYC, cells were first transduced with pRetroX-Tet-On Advanced, and selected for 7 days with 800 μg/mL geneticin (G418 Sulfate) (Thermo Fisher). Human MYC was PCR amplified from MSCV-Myc-T58A-puro, which was a kind gift from dr. Scott Lowe. MYC was digested with NotI and EcoRI and ligated into the corresponding cloning sites of pRetroX-Tight-Pur. Subsequently, cells were transduced with pRetroX-Tight-Pur harboring MYC and selected for two days with 5 μg/mL puromycin dihydrochloride (Sigma-Aldrich). To induce MYC expression, 1 μg/mL dox (Sigma-Aldrich) was added to the culture medium.

## Histology and immunohistochemistry
Tissues were formalin-fixed overnight and paraffin-embedded by routine procedures. Haematoxylin and eosin (HE) and immunohistochemical stainings were performed by standard protocols. The following primary rabbit antibodies were used for immunohistochemistry: anti-Myc (Abcam ab32072), anti-CD3 (Thermo Scientific, RM-9107), anti-F4/80 (abD serotec, MCA497), anti cleaved caspase-3 (cell signaling, #9661), anti Ki67 Abcam (ab15580) and anti-CD31 (AbCam ab28364). All slides were digitally processed using the Aperio ScanScope (Aperio, Vista, CA, USA) and captured using ImageScope software version 12.0.0 (Aperio) or quPath version 0.3.0[61].

## Generation of cGAS knockout cells
CRISPR guide RNAs were generated against cGAS (#1: 5′-cac-cgG-GCA-TTC-CGT-GCG-GAA-GCCT-3′; #2: 5′-cac-cgT-GAA-ACG-GAT-TCT-TCT-TTCG-3′) and cloned into the Cas9 plasmids pSpCas9(BB) − 2A-Puro (PX459, Addgene #62988) and pSpCas9(BB)−2A-GFP (PX458, Addgene #48138) using the AgeI and EcoRI restriction sites. BT-549 and HCC38 cells were transfected with both plasmids simultaneously (2 μg) using FuGene (Promega) according to the manufacturer's instructions. After transfection, cells were selected with puromycin (1 μg per mL) for 48 h or single-cell sorted for GFP. Single-cell *CGAS*$^{-/-}$ clones were confirmed by immunoblotting. Subsequently, *CGAS*$^{-/-}$ or parental cells were infected with Tet-pLKO-puro shRNAs targeting BRCA2 as described before.

## Western blotting
Cultured cells were lysed in Mammalian Protein Extraction Reagent (MPER, Thermo Scientific), supplemented with protease inhibitor and phosphatase inhibitor cocktail (Thermo Scientific). Proteins were separated on SDS-polyacrylamide gels (SDS-PAGE) and transferred onto a polyvinylidene difluoride (PVDF) membrane (Millipore). Membranes were blocked in 5% milk or bovine serum albumin (BSA) in Tris-buffered saline, with 0.05% Tween-20. Immunodetection was done with antibodies directed against BRCA2 (1:1000, Calbiochem, #OP95), BRCA1 (1:1000, Cell Signaling, #9010), cGAS (1:1000, Cell Signaling, #15102), STING (1:1000, Cell Signaling, #13647), cMYC (1:200, Santa Cruz, sc40, Abcam (ab32072) 1:1000), pIRF3 (1:1000, Cell Signaling, # 29047), IRF3 (1:1000, Cell Signaling, # 4302), STAT1 (1:1000, Cell Signaling, # 9172), pSTAT1 (1:1000, Cell Signaling, # 8826) and beta-Actin (1:10.000, MP Biochemicals, #69100). Horseradish peroxidase-conjugated secondary antibodies (1:2500, DAKO) were used and visualized with chemiluminescence (Lumi-Light, Roche diagnostics) on a Bio-Rad Bioluminescence device equipped with Quantity One/Chemidoc XRS software (Bio-Rad).

## In vitro survival assays
BT-549 or HCC38 cells with indicated hairpins were plated in 6 wells (500 cells per well) and treated with or without dox (1 μg per mL) for 10-14 days. Cells were fixed in methanol and stained with 0.1% crystal violet in H$_2$O. Plates were measured and quantified using an EliSpot reader (Alpha Diagnostics International) with vSpot Spectrum software. For proliferation assays, BT-549 and HCC38 cells with indicated hairpins were plated in 48 wells plates (10.000 cells per well) and cultured for up to 10 days with dox (1 μg per mL). At indicated time points, plates were centrifuged (900 RPM) for 10 minutes and cells were fixed with 10% Trichloroacetic acid (TCA) in H$_2$O overnight at 4 degrees. Plates were washed with tap water and dried by air. Cells were stained with 0.1% Sulforhodamine B (SRB) 1% Acetic acid in H$_2$O for 30 minutes at room temperature and subsequently washed with 1% Acetic acid-H$_2$O. Bound SRB dye was dissolved by adding 10 mM Tris-H2O to wells and OD was measured at 510 nM with an iMARK microplate reader (Bio-Rad).

## Quantitative RT-qPCR
Cell pellets from BT-549 and HCC38 treated with or without dox (1 μg per mL) for indicated time points were harvested and stored at −20 °C. RNA was isolated using the RNeasy Mini Kit (Qiagen) and complementary DNA (cDNA) was synthesized using SuperScript III (Invitrogen) according to the manufacturer's instructions. Quantitative reverse transcription-PCR (RT-PCR) for cytokine mRNA expression levels was performed in triplicate using PowerUp™ SYBR™ Green Master Mix (Thermo Scientific) or taqMan™ probes (for mouse *Ccl5* and *Cxcl10*) with TaqMan™ Fast Advanced Master Mix (Thermo Fisher Scientific). Glyceraldehyde 3-phosphate dehydrogenase (GAPDH), β-Actin or Rps22 was used as reference genes and experiments were performed on an Applied Biosystems Fast 7500 device.

## ELISA

To analyze cytokines and chemokines secreted by breast cancer cells, BT-549 and HCC38 cells with indicated hairpins were treated with or without dox (1 μg per mL) and plated at similar cell densities. Cell culture media was harvested at indicated time points and stored at −20 °C. Concentrations of CCL5 (R&D Systems, DY278-05) were measured using Enzyme-Linked Immuno Sorbent Assay (ELISA) according to manufacturer's instructions. ELISAs with supernatants of mouse organoid cultures for CXCL10 and CCL5 were performed according to the manufacturers instruction using the DuoSet ELISA system (R&D Systems, DY466 and DY478). Organoids were dissociated to single cells and plated at similar cell densities. Supernatants were collected 48 h after plating.

## Cytokine and chemokine array

According to the manufacturer's protocol, proteome profiler Mouse XL Cytokine array (R&D system) was performed on whole cell lysates from WB1P and WB1P organoids, according to manufacturer's protocol. Organoids were dissociated to single cells and plated at similar densities, and collected 48 h after plating.

## Immunofluorescence microscopy

Cells were grown on coverslips and treated with or without dox (1 μg per mL) for indicated time points. For RAD51 foci formation, cells were irradiated with 5 Gy using a CIS international/IBL 637 cesium[137] source. After 3 h of irradiation, cells were washed with PBS and fixed in 2% formaldehyde with 0.1% Triton X-100 in PBS for 30 min at room temperature. Cells were permeabilized in 0.5% Triton X-100 in PBS for 10 min and subsequently blocked with PBS containing 0.05% Tween-20 and 4% BSA for 1 h. For micronuclei staining, cells were fixed in 4% formaldehyde for 15 min at room temperature. Subsequently, cells were permeabilized with 0.1% Triton X-100 in PBS for 1 min followed by blocking in 0.05% Tween-20 and 2.5% BSA in PBS for 1 h. Cells were incubated overnight with primary antibodies against RAD51 (1:400, GeneTex, #gtx70230), Geminin (Cell Signaling, #9718, 1:200) or cGAS (1:200, Cell Signaling, #15102) in PBS–Tween–BSA. Cells were extensively washed and incubated for 1 h with Alexa-conjugated secondary antibodies (1:400) at room temperature in the dark. Slides were mounted with ProLong Diamond Antifade Mountant with DAPI (Thermo Scientific). Images were acquired on a Leica DM-6000RXA fluorescence microscope, equipped with Leica Application Suite software.

## ChIP-seq

Duplicate samples were used for ChIP-seq data generation. Organoids were cultured in 15 cm dishes. Medium was replaced by PBS containing 1% PFA and plates were left shaking for 10 min at RT.

Di(N-succinimidyl) glutarate (DSG) (2 mM) was then added and left shaking for 25 min after which reactions were quenched with 2.5 M glycine for 5 min. Organoids were then washed with ice-cold PBS+ protease inhibitor (Roche). ChIP and sample processing was performed as described previously[62]. Five μg of cMYC antibody Y69 (Abcam, ab32072) and 50 μl of magnetic protein A (10008D; Thermo Fisher Scientific) or 300 ul MIZ1 antibody (10E2) with 100ul Dynabeads were used per IP. For ChIP-seq of tumor tissue, OCT-embedded tumors were cut in 30um sections and processed as described[62]. The prepared libraries were sequenced with 65 base single reads on Illumina Hiseq 2500. The sequencing reads were aligned to the mouse genome GRCm38 (mm10) using Burrows-Wheeler Aligner (BWA, v0.7.5a[63]; with a mapping quality >20. Peak calling was performed using MACS2 v2.1.1.20160309 (q-value threshold 0.01, extension via Phantom Peaks). For each organoid and tumor dataset, the peaks from duplicate samples were merged based on the peak ranges using ChIPpeakAnno v3.18.2[64] and considered as MYC binding loci. The gene closest to each merged peak was defined as MYC target based on the GRCm38 (mm10)

genome annotation using ChIPpeakAnno. For co-occupancy analysis between MYC and MIZ1, MYC and MIZ1 ChIP-seq data were additionally generated in tumor samples ($n = 5$). The peaks for each sample were called by the same procedure as mentioned above. MYC and MIZ1 target peaks were defined as peaks detected in at least two samples and co-occupancies between MYC and MIZ1 were analyzed using DiffBind v.3.4[65]. Over-representation analysis for genes with promoters co-occupied by MYC and MIZ1 was performed by clusterProfiler[66]. Data deposited under ENA accession number "PRJEB43214".

## Flow cytometry

Tissues were collected in ice-cold PBS. Blood samples were collected in tubes containing heparin (Leo Pharma) and treated with red blood cell lysis buffer (155 mM $NH_4CL$, 12 mM $NaHCO_3$, 0.1 mM EDTA) (RBC). Tumors were mechanically chopped using a McIlwain Tissue Chopper (Mickle Laboratory Engineering) and digested either for 1 h at 37 °C in a digestion mix of 3 mg/ml collagenase type A (Roche, 11088793001) and 25 μg/ml DNAse (Invitrogen, 18068–015) or for 30 min at 37 °C in 100 μg/ml Liberase (Roche, 5401127001), in serum-free DMEM (Invitrogen). Reactions were terminated by addition of DMEM containing 8% FCS and cell suspensions were dispersed through a 70 μm cell strainer (BD Falcon, 352350). All single-cell suspensions were treated with RBC lysis buffer to remove red blood cells. Single-cell suspensions were plated in equal numbers in round bottom 96-wells plates (Thermo Scientific). Cells were incubated with mouse Fc Block™ (BD Biosciences) for 15 min at 4 °C and subsequently incubated with different combinations of fluorescently labeled monoclonal antibodies for 20 min in the dark at 4 °C.7AAD viability staining solution (eBioscience, 00–6993) was added to exclude dead cells. Flow cytometric analysis was performed on a BD LSRII using Diva Software (BD Biosciences). Data analyses were performed using FlowJo Software version 10.0 (Tree Star Inc.).

The following antibody panels were used: Myeloid panel – CD45-eFluor605NC (1:100; clone 30-F11, eBiosciences), CD11b-eFluor650NC (1:400; clone M1/70, eBiosciences), Ly6G-AlexaFluor700 (1:200; clone 1A8; BD Pharmingen), Ly6C-eFluor450 (1:400; clone HK1.4, eBiosciences), F4/80-PE (1:200; clone BM8, eBiosciences), CD49d-FITC (1:400; clone R1–2, eBiosciences), CD3 PerCP Cy5.5, CD206-FITC (1:200; clone C068C2, eBiosciences), 7-AAD (biolegend, cat. 420403); Lymphoid panel – CD45-eFluor605NC (1:50; clone 30-F11, eBiosciences), CD11b-eFluor650NC (1:400; clone M1/70, eBiosciences), CD3-PE-Cy7 (1:200; clone 145–2C11, eBiosciences), CD4-APC-eFluor450 (1:200; clone GK1.5, eBioscience), CD8-PerCP-eFluor710 (1:400; clone 53–6.7, eBiosciences, CD49b-APC (1:400; clone DX5, eBiosciences), CD19-eFluor780 (1:200; clone eBio1D3) 7-AAD.

For flow cytometry of mouse organoids and human cell lines BT-549 and HCC38, cells with indicated hairpins were cultured for different time points with dox and harvested by trypsinization and fixed with Fix buffer I (BD bioscience) for 30 min. on ice. Cells were washed with 1% BSA-PBS and permeabilized with Perm Buffer III (BD bioscience) for 30 min. on ice. Samples were washed with 1% BSA-PBS and incubated (150.000 cells per sample) with pIRF3 primary antibody (1:100, Cell signaling, #29047, clone D601M) for 1 h at 4 °C and subsequently stained with AlexaFluor 488-conjugated goat anti-rabbit secondary antibody (1:300) for 1 h at RT. Samples were measured on the FACS Calibur (Becton Dickinson), and data were analyzed using FlowJo software.

## RNA sequencing

TNBC cells with or without MYC overexpression were treated with dox (1 μg per mL) for 6 days. Cells were harvested and frozen at −80 °C. RNA was isolated using the mirVANA kit (Ambion, AM1561). To generate cDNA libraries suitable for next-generation sequencing (NGS), the QuantSeq RNAseq 3′ mRNA kit (Lexogen) was employed. The libraries were sequenced with 65 base-pair reads on a NextSeq

500 sequencer (Illumina), yielding 7.2 to 19.8 millions of reads per sample. FastQC and Samtools Flagstat software were used to assess RNA sequencing quality control. At least 80% of the bases had a Q-score ≥30. three biological replicates were used per cell line. RNA-seq data has been deposited at the GEO repository of the NCBI with identifier GSE185512.

For RNA sequencing of mouse tumors, RNA was isolated from tumor pieces with the Qiagen RNA isolation kit. The mRNA library was generated using Illumina TrueSeq Stranded mRNA Library Prep Kit and sequenced with 65 base single-end reads on Illumina Hiseq 2500. The sequencing reads were aligned to the mouse genome GRCm38 (mm10) using TopHat v2.1[67] and the number of reads mapped to each gene was quantified using HTSeq[68]. DESeq2 v1.22.2 was used for read count normalization (median ratio method) and differential expression analysis. Genes with adjusted FDR < 0.05 (Benjamini-Hochberg procedure) and |fold-changes |>1.5 were defined as differentially expressed genes. Data deposited under ENA accession number "PRJEB43214".

### Trans-well T cell migration assay

BT-549 and HCC38 cells with indicated shRNAs were plated in 24-well plates (20,000 cells per well) and treated with dox (1 μg per mL) for 4 or 5 days. Human peripheral blood mononuclear cells (PBMCs) were isolated from peripheral blood from healthy volunteers (buffy coat obtained from Sanquin, Amsterdam, the Netherlands) by Ficoll-Paque density centrifugation (Ficoll-Paque PLUS, GE Healthcare Life Sciences) and enriched for CD8+ T cells with the MagniSort™ Human CD8+ T cell Enrichment Kit (#8804-6812-74, Invitrogen) according to manufacturer's instructions. Enriched CD8+ T cells (750,000 cells per transwell) were added on top of the filter membrane of a transwell insert (6.5 mm Transwell with 3.0 μm pore, Corning) and incubated for 24, after which supernatant from the lower chamber was harvested to quantify migrated T cells by microscopy.

### T cell proliferation assay

BT-549 and HCC38 cells were plated in 6-well plates (20,000 per well) and treated with dox (1 μg per mL) for 5 days. T cells were harvested and enriched for CD8+ T cells as described for the T cell migration assay. Enriched CD8+ T cells were stained with CellTrace Violet (#C34557, ThermoFisher) according to manufacturer's instructions and cultured in 96-well plates (100,000 cells per well) with 200 μL conditioned medium harvested from breast cancer cells pre-treated with dox for 5 days. To activate T cells, T cells were co-cultured with Human T-Activator CD3/CD28 dynabeads (#11131D, Thermofisher) in a bead to T cell ratio of 1:4 or 1:8. For every condition, 2 wells were cultured and combined for analysis. At day of analysis, T cells were pooled, harvested, measured on the FacsVerse (BD Biosciences) and analyzed with FlowJo software.

### Organoid-splenocyte co-culture

Organoids were derived from WB1P or WB1P-Myc mammary tumors as described[69]. WB1P organoids were transduced with a lenti-GFP and WB1P-Myc with a lenti-mCherry lentivirus. Splenocytes were derived from FVB mouse spleen, by dissociation on a 70uM cell strainer. For the co-culture, 200,000 splenocytes and 10 organoids were plated together in a 24-well plate with 50% RPMI medium, 50% ENR medium, supplemented with IL-2 (Prepotech, 300IU/ml). Live cell imaging was performed with a Zeiss AxioObserver Z1 microscope for 7 days. Organoid areas were quantified using Zen software.

For the MTT assays, roughly 1,000 cells, disrupted by a fire-hardened glass pipette to approximately 5-10 cells/clump were seeded together with 20000 splenocytes in a 96 well plate, using the same culturing conditions as described above. Vadimezan (MedKoo, #201050) was added at 10ug/ml for 7 days, then 3-(4,5-Dimethylthiazol-2-yl)−2,5-Diphenyltetrazolium Bromide, (ThermoFisher, #M6494) was added for 3 h, followed by cell lysis for 16 h in SDS lysis buffer.

Plates were analyzed on a TECAN infinite M-plex plate reader. For co-culture with CD8+ T cells, T cells were isolated from mouse splenocytes using a CD8a+ T cell isolation kit (#130-104-075, Miltenyi Biotec) and LS columns (#130-042-401, Miltenyi Biotec). Enriched CD8+ T cells were activated with Dynabeads Mouse T-activator CD3/CD28 (#11456D, Thermofischer) in a bead to T cell ratio of 1:5. 30,000 CD8+ T cells and 50 dissociated organoids were plated together in a 24-well plate with 50% splenocyte medium (IMDM+ Glutamax + HEPES+ P/S+ FCS+ β-mercaptoethanol) and 50% organoid medium supplemented with IL-2 (300IU/ml). Organoids were imaged with an Invitrogen EVOS FL microscope after 1, 4 and 7 days of co-culture. Organoid areas were quantified using ImageJ software.

### TCGA data preprocessing and quality control

Genes with a robust average gene expression (Hodges Lehmann estimate) lower than 20, were removed from the analysis. Differences in gene expression due to differences in cancer types were adjusted for every cancer type separately by performing the following steps for each gene: (i) robust average gene expression was obtained using Hodges Lehmann estimator; (ii) robust standard deviation of gene expressions were obtained using Hall's estimator; (iii) gene expression was normalized using the following formula: Adjusted gene expression = (gene expression − robust average)/robust standard deviation.

### Analysis of TONIC trial RNA-seq data

Pre-processed RNA-seq data for pre-treatment samples from TONIC trial were obtained upon data request (EGAS0001003535). The raw data was pre-processed by following steps: 1) gene-specific read counts for the Ensembl version 86 build of the human transcriptome on reference genome GRCh38 were obtained by running Salmon v0.11.0[70] directly on the FASTQ files using default settings. 2) Transcript-specific read counts were collapsed to gene expression read counts using the R Bioconductor package tximport v1.4.0[71]. 3) Read counts were subsequently trimmed mean of M values (TMM)-normalized using the edgeR Bioconductor package v3.18.1. Pearson's correlation was computed between MYC expression and all other protein-coding genes and the resulting correlation coefficients were used for GSEA analysis by fgsea Bioconductor package v1.8.0[72].

### Differential gene expression analysis

To investigate the differential gene expression in the context of amplification of oncogenes, we retrieved DNA copy number data from The Cancer Genome Atlas (TCGA). For each of the oncogenes, the respective copy number profiles were used to classify samples as either amplified ($\log_2$(segment mean copy number) > 0.3) or neutral ($0.3 \geq \log_2$(segment mean copy number) ≥ −0.3). After that, Welch t-test was performed to identify differentially expressed genes upon amplification of each oncogene. A metric defined by (−log10(p-value) *sign(t statistic)) for each Welch t-test was obtained. The above analysis was done separately on the following sets of samples from TCGA: (i) all breast cancer samples; (ii) TNBC samples; (iii) breast cancer samples with high (mutation assumed to have a disruptive impact on protein) or moderate (mutation possibly changing protein effectiveness) mutation in either BRCA1 or BRCA2; (iv) TNBC samples with a high or moderate disruptive mutation in either BRCA1 or BRCA2.

### Gene Set Enrichment Analysis (GSEA)

For GSEA of oncogene-expressing and control BT-549 or HCC38 cells, genes were ranked based on the −log P value between oncogene-expressing cells and control cells (pBABE-empty). Genes enriched in oncogene-expressing cells were positive and genes enriched in control cells were negative. For GSEA of BRCA2-depleted cells with or without MYC overexpression, genes were ranked based on the -log P-value between MYC overexpressing cells and control cells. Genes enriched in MYC-overexpressing cells were positive and genes in control cells were

negative. Gene sets of the Hallmark collection (MSigDB) were loaded into GSEA and analyzed in both cell lines. For GSEA of BRCA2-depleted cells with or without MYC overexpression, only significantly down-regulated genes (p < 0.05) in MYC overexpressing cells were loaded into GSEA software for both cell lines. GSEA was performed utilizing 3 gene set databases (Hallmark, Reactome & Gene Ontology Biological Processes) from the MSigDB.v5.2[73]. Gene sets containing less than 10 genes or more than 500 genes (after filtering out genes that were not present in our data sets) were excluded from further analysis. Enrichment of a gene set was tested according to the two-sample Welch's t-test for unequal variance. Welch's t-test was conducted between the set of metrics obtained from differential gene expression analysis of genes whose corresponding gene identifiers are members of the gene set under investigation and metrics of genes whose corresponding gene identifiers are not members of the gene set under investigation. To compare gene sets of different sizes, Welch's t statistics were transformed to -log10(P-value).

GSEA of mouse mammary tumors was performed based on the Wald statistic obtained from DESeq2 differential expression analysis using the fgsea Bioconductor package v1.8.0[72]. MsigDB Hallmark gene sets[73] with a minimum size of 15 and a maximum size of 3000 were used for enrichment analysis.

## Differential immune cell type abundance

Immune cell type abundance in breast cancer samples from TCGA was estimated using CIBERSORT[34]. The abundance of 22 immune cell types was estimated by applying the leukocyte gene signature matrix (LM22) on the mRNA expression profiles from TCGA. To investigate the differential immune cell type abundance in the context of amplification of MYC, we used DNA copy number data from TCGA to classify samples as either MYC-amplified ($\log_2$(segment mean copy number) > 0.3) or neutral ($0.3 \geq \log_2$(segment mean copy number) $\geq -0.3$). After that, Welch t-test was performed to identify immune cell types that showed statistically significantly different abundance in MYC amplified versus neutral samples. A metric defined by ($-\log10(p$-value)*sign(t statistic)) for each Welch t-test was obtained to explore the result. The above analysis was done separately on the following set of samples from TCGA: (i) all breast cancer samples; (ii) TNBC samples.

## Prediction of gene functionalities

A co-functionality network was generated with an integrative tool that predicts gene functions based on a guilt-by-association (GBA) strategy utilizing >106,000 expression profiles as described previously (Bhattacharya et al.[41]). The analyzer tool is available at http://www.genetica-network.com.

## Reporting summary

Further information on research design is available in the Nature Research Reporting Summary linked to this article.

## Data availability

Mouse RNA-seq and ChIP-seq data has been deposited with the ENA accession number "PRJEB43214" RNA-seq presented in Fig. 1 has been deposited with GEO repository with identifier: "GSE185512" Figures with associated raw data: Figures 1–3, 5, Supplementary Figs. S1–S3, S7, S8. Source data are provided with this paper.

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

## Acknowledgements

We are grateful for excellent support from the NKI animal facility, RHPC computing facility, flow cytometry facility, animal pathology facility, transgenic facility, preclinical intervention unit, core facility molecular pathology and biobanking (CFMPB), and genomics core facility. We want to thank Martine van Miltenburg for providing data regarding the KP tumor model and Ivo Huijbers for help with generating the mouse models. This work was carried out on the Dutch national e-infrastructure with the support of SURF Cooperative (e-infra160136). Financial support was provided by the Oncode Institute, the Netherlands Organization for Scientific Research (NWO: Cancer Genomics Netherlands (CGCNL), VICI 91814643 (J.J.), VICI 91819616 (K.E.d.V.), VIDI 91713334 (M.A.T.M.v.V.)), the European Research Council (ERC Synergy project CombatCancer (J.J.), ERC-Consolidator grant "TENSION" to (M.A.T.M.v.V.), ERC- Consolidator award InflaMet 615300 (K.E.d.V.)), a National Roadmap grant for Large-Scale Research Facilities from NWO (J.J.), the Dutch Cancer Society KWF13191 and KWF10623 (K.E.d.V.), an unrestricted grant of the Hanarth Fonds (R.S.N.F) and a postdoc mobility grant from the Swiss National Science foundation (D.Z).

## Author contributions

D.Z.: conceptualization, investigation, data analysis, writing of original draft, review and editing C.B.: conceptualization, investigation, data analysis, writing of original draft, review and editing F.T.: conceptualization, investigation, data analysis, writing of original draft, review and editing J.B.: Data analysis L.R.: Methodology, data analysis A.B.: Data analysis S.J.: Methodology, data analysis A.M.: Methodology, data analysis N.P..: Methodology R.L.: Methodology, data analysis M.D.W: Methodology, data analysis K.K.: Methodology, data analysis M.R.: Data analysis L.H.: Methodology, data analysis R.dB.: Data analysis S.A.: Methodology E.vdB.: Methodology AP.D.: Methodology C.L.: Methodology T.E.: Methodology M.vdW.: Supervision M.E.: Supervision L.W.: Supervision K.E.dV.: Supervision W.Z.: Supervision R.S.N.F.: Supervision, review & editing, Methodology, data analyses, and Conceptualization M.A.T.M.vV: Conceptualization, supervision, writing of original draft, review & editing J.J: Conceptualization, supervision, writing of original draft, review & editing.

## Competing interests

K.E.d.V. receives research funding from Roche/Genentech and is consultant for Macomics, both outside the scope of this manuscript. The other authors declare no potential conflict of interest.

## Additional information

¹Division of Molecular Pathology, The Netherlands Cancer Institute, Amsterdam, The Netherlands. ²Oncode Institute, Utrecht, The Netherlands. ³Department of Medical Oncology, University Medical Center Groningen, University of Groningen, Groningen, the Netherlands. ⁴Division of Molecular Carcinogenesis, The Netherlands Cancer Institute, Amsterdam, The Netherlands. ⁵Translational Research Centre in Oncohaematology, University of Geneva, Geneva, Switzerland. ⁶Division of Oncogenomics, The Netherlands Cancer Institute, Amsterdam, The Netherlands. ⁷Division of Tumor Biology & Immunology, The Netherlands Cancer Institute, Amsterdam, The Netherlands. ⁸Transgenic Core Facility, Mouse Clinic for Cancer and Aging (MCCA), The Netherlands Cancer Institute, Amsterdam, The Netherlands. ⁹Theodor Boveri Institute, Department of Biochemistry and Molecular Biology, Biocenter, University of Würzburg, Am Hubland, 97074 Würzburg, Germany. ¹⁰Preclinical Intervention Unit, Mouse Clinic for Cancer and Aging (MCCA), The Netherlands Cancer Institute, Amsterdam, The Netherlands. ¹¹Present address: Hubrecht Institute–KNAW (Royal Netherlands Academy of Arts and Sciences) and University Medical Center Utrecht, Utrecht, The Netherlands. ¹²Present address: Department of Pathology, University of California, San Francisco, CA, USA. ¹³Present address: Chemical and Systems Biology, Stanford Uiversity School of Medicine, Stanford, USA. ¹⁴These authors contributed equally: Dario Zimmerli, Chiara S. Brambillasca, Francien Talens. ✉e-mail: r.s.n.fehrmann@umcg.nl; m.vugt@umcg.nl; j.jonkers@nki.nl

