## [Peer Review File · Nature Communications]

MYC promotes immune-suppression in triple-negative breast cancer via inhibition of interferon signalingREVIEWER COMMENTS

Reviewer #1 (Breast cancer, tumor microenvironment.) (Remarks to the Author):

In this manuscript, Zimmerli, et al. reported a critical role for Myc in suppressing anti-tumor immunity in triple negative breast cancer. The authors first observed MYC expression is negatively correlated with IFN and JAK-STAT signaling in human TNBC datasets. Using RNAseq and ChIP-seq, the authors revealed MYC downregulates interferon signaling genes by directly binding to their promoter regions in mouse TNBC organoids and tumors. Further study confirmed that Myc overexpression reduces BRCA1/2 depletion induced CCL5 secretion and this inhibition was achieved through cGAS/STING pathway. In vitro functional assay showed that Myc overexpression blocks CD8+ T cell migration, proliferation and activation through tumor cell secreted factors. By overexpressing MYC in the mouse TNBC models, the authors observed significant reduction of tumor infiltrating lymphocytes and myeloid cells, and this immune-depleting effects was more profound in the BRCA1 deficient tumors.

Overall, the authors provided strong evidence supporting that MYC overexpression promotes immune suppression in the TNBC tumors through inhibiting IFN signaling. However, the novelty of this manuscript is somewhat compromised by published work in other tumor models that reach similar conclusions. Topper, et al. reported MYC downregulates CCL5 expression and antigen presentation related molecules in a type I IFN signaling dependent mechanism in NSCLC model. MYC targeting treatment could reverse tumor immune evasion in mouse NSCLC model (Topper, et al. *Cell* 2017; 171(6): 1284-1300). There are some specific issues the authors need to address.

1. Based on method section, mice were sacrificed upon tumor reaching 1500 mm³. Is it a fair comparison to analyze the tumor infiltrating immune cells if it takes much longer for the control tumors to reach the same size as the MYC overexpression tumors?
2. Figure 2B, the authors quantified tumor infiltrating CD3+ cells in four tumor models. Similar quantification plots in Figure 4F and 4I used different y axis label. It is not clear how the analysis was performed. The authors need to explain this in detail in the method section or figure legend.
3. To exclude the possibility that reduced immune cell infiltration was due to accelerated tumorigenesis in the MYC overexpressing model, the authors used a Met overexpressing model as a control. MYC functions as a transcription factor and primarily localizes to the nucleus of cells. However, MET is a receptor tyrosine kinase which is potentially more immunogenic than MYC. The reviewer suggests using a nucleus or cytoplasm localized oncoprotein as a control.
4. In the flow cytometry method section, the authors used detailed antibodies that can investigate specific subsets of immune cell populations but did not incorporate the data with the same granularity in the figures. For example, they used Ly6C, Ly6G and F4/80 antibodies in the myeloid panel, but only showed 'myeloid cells' in Figure 2D.
5. Figure 3B-D and Figure S4B, the authors used qRT-PCR to quantify Cxcl10 expression, cytokine array to quantify CCL5 secretion and western blot to measure STAT1 protein level. It is better to quantify these factors in a consistent manner, at least for the same type of factors (e.g., chemokines).
6. Figure 4B, it seems that the second bar from the left have only two data point, therefore it should not be used for statistical analysis. Same issue happens to Figure 4E, the left most group have only two data points. Figure 4C, none of the comparisons reached statistic significance, the authors cannot draw a conclusion based on these data.
7. In Figure 4, the authors revealed that MYC-expressing cancer cells modulate lymphocytes activity, especially CD8+ T cell proliferation and activation. Figure 6C, the authors co-cultured organoid with splenocytes to test MYC driven immune-suppressive activities. The reviewer suggests co-culture organoids with selected CD8+T to exclude the confounding effects from other splenocytes population.

Reviewer #2 (Myc in cancer.) (Remarks to the Author):

Using data generated from a wide range of in vitro assays and multiple clinically relevant transgenic mouse models for triple negative breast cancer, Zimmerli et al. have shown that MYC is a critical factor that suppresses innate immunity and promotes immune escape in triple negative breast cancer. These findings will significantly enhance our current understanding of the

mechanisms underlying the immunosuppressive phenotype displayed by TNBCs and BRCA-mutated breast cancers, especially those with MYC-overexpression. However, the data elucidating the paracrine effects of tumor cells on TILs could be further strengthened to help answer questions such as how inhibition of the interferon pathway in the tumour cell suppresses immune-cell infiltration and what paracrine factors and signals are involved. Please see more comments below.

Major points:

1. It seems that the major tumor infiltrating lymphocyte (TIL) population in WB1P tumors is CD11+ myeloid cells, and the percentage decrease in these CD11+ myeloid cells in WB1P-MYC tumors is almost identical to that in the total CD45+ cells, suggesting that the majority of the TILs expelled by MYC expression are CD11b+ cells (Figure 2D). Have the authors further characterized these CD11b+ cells since many of the cell types (such as monocytes, NKs, macrophages, etc.) play important roles in the interferon response?
2. As quantifications of immune cells by flow are based on the percentage of live cells, comparison of tumor cell death in the WP, WP-MYC, WB1P, and WB1P-MYC tumor models by either staining or flow should be included to show that the decrease in immune cell populations is not due to changes in percentages of live tumor cells.
3. Do WP and WP-MYC tumors progress at a different pace, as tumor size may impact immune cell infiltration (Kim et al, *Front. Immunol.*, 2021 <https://doi.org/10.3389/fimmu.2020.629722>)? Were these tumors of similar sizes when collected? It would also strengthen the data to include a comparison of immune cell infiltration in tumors of similar sizes from both early and late stages of tumor development in these models.
4. The authors have shown rather convincingly that MYC overexpression in tumor cells inhibits immune-cell infiltration in the TME. However, I find that the link between suppression of the STING pathway by MYC in tumor cells and its downstream paracrine effects on immune cells is not evident. Were the decreased TIL levels in MYC-overexpressing tumors due to suppression of the proinflammatory factors (such as CXCL10, CCL5, and interferons) or elevated immunosuppressive molecules secreted by the tumor cells? Can the authors measure changes in levels of cytokines and chemokines in MYC-overexpressing tumors (and/or in serum) vs in non-MYC-expressing tumors (and/or in serum)? In the in vitro co-culturing assays, would addition of some of these decreased cytokines rescue T-cell killing, T-cell migration, and T-cell proliferation in the presence of MYC-overexpressing tumor cells/organoids?
5. MYC-overexpressing tumors are normally more aggressive than non-MYC-overexpressing tumors, so it is expected that these tumors progress faster than non-MYC tumors as shown by the authors in several different mouse models. Although the authors have shown that MYC-overexpressing organoids were more resistant to T cell killing in vitro, this is not shown in vivo, and it is not clear if MYC-inhibited immune infiltration actually enhance tumor progression or simply render the tumors more resistant to immunotherapy. I wonder if the authors have further delineated these relationships. In vadimezan-treated animals in Figure 6, did authors observe an increased immune infiltration in WB1P and WB1P-MYC tumors? Do levels of immune infiltration correlate with survival or tumor growth?

Minor points:

1. Scale bars need to be added to Figures 2C, 2E, 4D, S3B, S6A, S6B, and S6D.
2. In Figure 2C, WB1P and WB1P-MYC are mislabeled as WP1P and WP1P-MYC.
3. Make sure the labels for figure axes are consistent. For example, in graphs for CD3 IHC quantification, the Y-axis in some figures is labelled as "CD3+ (average # of cells)", whereas others as "CD3+/area" or "CD3+ cells (ave number/area)".
4. In S3B, the Lenti-sgPTEN-Myc tumor image appears to be very pixelated compared with the Lenti-sgPTEN image. Do they have the same magnification?
5. Why are different types of statistical tests (t-test, Mann-Whitney tests) used to determine

statistical significance? The type of statistical test is not stated in figure legends for Figures 4B and 4D. Figure 3H does not have statistical analysis. Some of the statistical tests are also not used correctly. For example, in Figures 3G, 4B, 4D, 4E, and 4I, one-way or two-way ANOVA tests should be used accordingly instead of unpaired t-tests since these datasets contain multiple (>3) groups for comparison (i.e., one-way ANOVA with post hoc multiple comparisons for Figures 4E and 4I, and two-way ANOVA with post hoc multiple comparisons for Figures 3G, 4B, and 4D where two variables are included).

6. Is it possible to include representative images containing multiple cells and images of control cells in Figure 3F?

7. In S4F, were the shLUC transduced BT549 cells in the right panel treated with dox? If so, "+" and/or "-" symbols will need to be added. The middle panel images seem to contain protein bands cut and pasted from different blots – please clarify.

8. Survival curves of WB1P-MYC mice should also be added to Figure 6A.

9. In Figure 6B, perhaps a simpler and more conventional dot plots similar to Figure 6C, rather than the current waterfall plot, could be used to show changes in tumor volumes.

10. In Figure 6D, the flame symbol with a cross is a bit confusing especially when it is positioned on the cell membrane. As readers may not necessarily understand a flame symbol as inflammation, perhaps include some text to explain it and connect it with the rest of the pathway? In addition, this illustration only shows what occurs within the tumor cell. How these changes impact antitumor immunity should also be demonstrated.

Reviewer #3 (cGAS/STING in cancer) (Remarks to the Author):

In this paper 'MYC promotes immune-suppression in TNBC via inhibition of IFN signaling' by Zimmerli et al., the authors surveyed the TNBC samples within the TCGA database and the TONIC phase II trial and found that MYC expression is associated with IFN and inflammatory gene downregulation. Using MYC-overexpressing human TNBC cell lines and mouse TNBC models, they further showed that MYC overexpression drives immune cell tumor exclusion by suppressing expression of IFN-stimulated genes (ISGs) and inflammatory genes in a tumor-cell intrinsic manner. With CHIP-seq analysis, the authors further showed that MYC serves as a master transcriptional regulator by directly binding to the promoter regions of a myriad of IFN and inflammatory pathway genes. Overall, the study provides an important mechanism by which TNBCs and BRCA-mutated breast cancers may achieve tumor immune evasion and respond poorly to ICI. This is an interesting paper with a novel finding about the effect of MYC on tumor immunity. We appreciate the various GEMM models, organoids, and human TNBC cell lines used and the detailed bioinformatic analyses the authors performed to support their findings. However, there are few issues the authors may need to address before making a more concrete conclusion.

Major points:

1. The authors relied on overexpressing MYC in human TNBC cell lines and mouse TNBC models to study the effect of MYC. It remains unknown what the endogenous level of MYC expression is in these models, and what is the effect of endogenous MYC on immune gene transcription and immune cell tumor recruitment. It would be more convincing to add data with MYC knockout tumor models, and to compare MYC expression in early vs late stage TNBC models and correlate endogenous MYC expression with tumor immune signature and gene expression.

2. The authors used CHIP-seq to show that MYC functions as a master regulator by targeting numerous IFN and inflammatory signaling genes through binding to their promoter region and downregulating target gene expression. However, no mechanistic insight was provided to explain how MYC achieved this transcriptional inhibition. It would be great if the authors could offer some mechanistic evidence to support their finding or at least discuss what they are thinking.

3. Many experiments have very small sample size ($n \leq 4$), for example Figures. 3D, 4B, and 4C; or do not show statistical analysis, for example Figures. 3H, S5A, and 6B, making it hard to determine if these data support the authors' conclusions.

Minor point:

1. When analyzing numbers of tumor-infiltrating CD3+ cells in Figures. 2C and S3B, the authors did not specify the time of analysis following tumor induction. It would be more informative to isolate TILs at different time points and compare when Myc starts to affect TIL number in the tumor. In addition, it will also be important to determine whether TIL functions were impaired with Myc expression.

In S3B, the CD3+ cell number in WB1P-Cas9 control tumors is missing (without lentiviral injection). The data from controls will help determine whether PTEN loss on its own has already reduced CD3+ TIL numbers.

2. In Figures 3E and S4D, the authors analyzed pIRF3 levels in the whole tumor. It will be more informative to gate on tumor cells to determine if the effect of MYC on IRF3 phosphorylation is tumor cell-specific.

3. Figure. 6A did not show effect of vadimezan on progression-free survival of WB1P-Myc tumors.

5. In Figure. 6B, it is hard to tell the efficacy of vadimezan with this waterfall plot. It would be better to compare tumor growth kinetics and include statistical analysis.

6. In Figure. 6C, it is expected that organoid co-culture with splenocytes should inhibit tumor growth (as shown in Figure. 4D), making MTT conversion ratios < 1 . It is hard to understand why the MTT conversion ratios in vehicle treated WB1P or WB1P-MYC tumors are > 1 .

7. Typos, Figure. 2C labels should be B1P instead of P1P. Sup Fig. S5F should be S5E.

Reviewer #1:

In this manuscript, Zimmerli, et al. reported a critical role for Myc in suppressing anti-tumor immunity in triple negative breast cancer. The authors first observed MYC expression is negatively correlated with IFN and JAK-STAT signaling in human TNBC datasets. Using RNAseq and ChIP-seq, the authors revealed MYC downregulates interferon signaling genes by directly binding to their promoter regions in mouse TNBC organoids and tumors. Further study confirmed that Myc overexpression reduces BRCA1/2 depletion induced CCL5 secretion and this inhibition was achieved through cGAS/STING pathway. In vitro functional assay showed that Myc overexpression blocks CD8+ T cell migration, proliferation and activation through tumor cell secreted factors. By overexpressing MYC in the mouse TNBC models, the authors observed significant reduction of tumor infiltrating lymphocytes and myeloid cells, and this immune-depleting effects was more profound in the BRCA1 deficient tumors.

Overall, the authors provided strong evidence supporting that MYC overexpression promotes immune suppression in the TNBC tumors through inhibiting IFN signaling. However, the novelty of this manuscript is somewhat compromised by published work in other tumor models that reach similar conclusions. Topper, et al. reported MYC downregulates CCL5 expression and antigen presentation related molecules in a type I IFN signaling dependent mechanism in NSCLC model. MYC targeting treatment could reverse tumor immune evasion in mouse NSCLC model (Topper, et al. Cell 2017; 171(6): 1284-1300). There are some specific issues the authors need to address.

Comment 1. *Based on method section, mice were sacrificed upon tumor reaching 1500 mm³. Is it a fair comparison to analyze the tumor infiltrating immune cells if it takes much longer for the control tumors to reach the same size as the MYC overexpression tumors?*

Reply: While the median latency until detection of the tumor is indeed much shorter for the WB1P-Myc model, there is no significant difference in growth rate once a tumor is detectable (revised Supplementary Figure S2F). We therefore think it is indeed fair to compare the WB1P and WB1P-Myc models.

We also added quantifications of Ki67 and Caspase-3 to check the proliferation and apoptosis rates (revised Supplementary Figure S2G,H). We observe that while WB1P-Myc tumor cells do divide faster, also the apoptosis rate is higher, explaining the similar overall growth rates.

Furthermore, we also used the WB1P-MycERT2 model to induce and de-induce MYC function in established tumors and observed the same phenotype of immune evasion as in WB1P-Myc tumors, showing it is independent of the underlying genotype and latency (revised Figure 4F-I).

Comment 2. *Figure 2B, the authors quantified tumor infiltrating CD3+ cells in four tumor models. Similar quantification plots in Figure 4F and 4I used different y axis label. It is not clear how the analysis was performed. The authors need to explain this in detail in the method section or figure legend.*

Reply: We thank the reviewer for pointing this out. We corrected the labels of the graphs and updated the methods section. We indeed counted CD3+ cells in 5 defined areas for each tumor in these figures, so the label should be "average number of CD3+ cells/Area".

Comment 3. *To exclude the possibility that reduced immune cell infiltration was due to accelerated tumorigenesis in the MYC overexpressing model, the authors used a Met overexpressing model as a control. MYC functions as a transcription factor and primarily localizes to the nucleus of cells. However, MET is a receptor tyrosine kinase which is potentially more immunogenic than MYC. The reviewer suggests using a nucleus or cytoplasm localized oncoprotein as a control.*

Reply: This is indeed a good point in principle, but we do not think this is an issue in this case, because we are using wild-type murine MET, which will not be targeted by the mouse immune system. Furthermore, MET expression is induced by WapCre, which was shown to be expressed early enough in order to induce immune tolerance to the overexpressed proteins; even to CAS9, which is known to be immunogenic in mice (Annunziato *et al.*, Genes Dev 2016;30:1470-80). We also added a panel in revised Supplementary Figure S3B showing the immune infiltration in untreated WB1P-Cas9 animals (CAS9 being expressed upon WapCre mediated recombination) in comparison to WB1P-Cas9 animals injected with lentivirus encoding MYC and a guide against PTEN. Also in this setup, we see very strong reduction of immune infiltration upon MYC overexpression. Another reason why we think it highly unlikely that MET would be more immunogenic than MYC, is the fact that both oncogenes are co-expressed with luciferase from a single bicistronic mRNA transcript. Overexpression of luciferase should cause a much stronger immune reaction than overexpression of wild-type murine MET, if expression via the WapCre system would be insufficient to render the mice tolerant towards the overexpressed proteins.

Furthermore, we used intraductal injections of lentiviruses encoding either MYC or MYC-V394D into B1P mice to generate mammary tumors. These experiments confirmed the dependency of immune expulsion on functional MYC-mediated transcriptional repression, since the MYC mutant MYC-V394D (which is incapable of binding the MYC co-repressor MIZ1 (van Riggelen *et al.*, Genes Dev 2010;24:1281-94)) was not able to expel immune cells to the same level as wild-type MYC (revised supplementary Figure 5D).

Comment 4. *In the flow cytometry method section, the authors used detailed antibodies that can investigate specific subsets of immune cell populations but did not incorporate the data with the same granularity in the figures. For example, they used Ly6C, Ly6G and F4/80 antibodies in the myeloid panel, but only showed 'myeloid cells' in Figure 2D.*

Reply: We thank the reviewer for pointing this out. We added the FACS data of the more specific myeloid panel (revised Figure 2D), where we see a strong reduction of immune infiltrating cells regardless of the population we assess.

Comment 5. *Figure 3B-D and Figure S4B, the authors used qRT-PCR to quantify Cxcl10 expression, cytokine array to quantify CCL5 secretion and western blot to measure STAT1 protein level. It is better to quantify these factors in a consistent manner, at least for the same type of factors (e.g., chemokines).*

Reply: We performed qRT-qPCRs as well as ELISAs for CCL5 and CXCL10 to unify this analysis (revised Figure 3B-D). We furthermore included the data from the whole cytokine array we performed (revised Supplementary Figure S4C). For pSTAT1, western blot analysis was the most sensitive and quantitative method.

Comment 6. *Figure 4B, it seems that the second bar from the left have only two data point, therefore it should not be used for statistical analysis. Same issue happens to Figure 4E, the left most group have only two data points. Figure 4C, none of the comparisons reached statistical significance, the authors cannot draw a conclusion based on these data.*

Reply: We agree with the reviewer. To increase the number of data points, we repeated the transwell assays with human BT549 breast cancer cells, which are statistically significant at the 24h time-point for BRCA2 and confirm our hypothesis (revised Figure 4B). Upon BRCA1 depletion, we observed a trend towards increased cytokine expression, which was reversed upon MYC overexpression, but these analyses did not reach statistical significance, most likely due to incomplete knock-down of BRCA1 as shown in the Western Blots in Supplementary Figures S4F and S4J. We also removed the 'T cells alone' data from the FACS analysis of the WB1P and WB1P-Myc organoids co-cultured with T cells and stained for CFSE, since these data are not relevant for our comparison of WB1P versus WB1P-Myc organoids (revised Figure 4D).

Furthermore, we repeated the experiments for T cell activation (previous Figure 4C), but they did not reach statistical significance, although we do see a trend in line with T cell migration. We conclude that T cell activation is not strongly hampered by MYC overexpression in human BT549 cells with BRCA1/2 depletion. We have changed the text in the revised manuscript and moved the data to revised Supplementary Figure S5C.

Comment 7. *In Figure 4, the authors revealed that MYC-expressing cancer cells modulate lymphocytes activity, especially CD8+ T cell proliferation and activation. Figure 6C, the authors co-cultured organoid with splenocytes to test MYC driven immune-suppressive activities. The reviewer suggests co-culture organoids with selected CD8+T to exclude the confounding effects from other splenocytes population.*

Reply: The reviewer is correct that there might be confounding effects from other splenocytes. We therefore repeated the organoid co-culture experiments with CD8⁺ T cells obtained by magnetic bead sorting. These experiments confirmed the observations from the organoid co-culture experiments with splenocytes, i.e., that WB1P organoids are more susceptible to immune cell killing than WB1P-Myc organoids, even though CD8⁺ T cells are overall more potent than splenocytes (revised Figure 4C,E).

Reviewer #2:

Using data generated from a wide range of in vitro assays and multiple clinically relevant transgenic mouse models for triple negative breast cancer, Zimmerli et al. have shown that MYC is a critical factor that suppresses innate immunity and promotes immune escape in triple negative breast cancer. These findings will significantly enhance our current understanding of the mechanisms underlying the immunosuppressive phenotype displayed by TNBCs and BRCA-mutated breast cancers, especially those with MYC-overexpression. However, the data elucidating the paracrine effects of tumor cells on TILs could be further strengthened to help answer questions such as how inhibition of the interferon pathway in the tumour cell suppresses immune-cell infiltration and what paracrine factors and signals are involved. Please see more comments below.

Major points:

Comment 1. *It seems that the major tumor infiltrating lymphocyte (TIL) population in WB1P tumors is CD11+ myeloid cells, and the percentage decrease in these CD11+ myeloid cells in WB1P-MYC tumors is almost identical to that in the total CD45+ cells, suggesting that the majority of the TILs expelled by MYC expression are CD11b+ cells (Figure 2D). Have the authors further characterized these CD11b+ cells since many of the cell types (such as monocytes, NKs, macrophages, etc.) play important roles in the interferon response?*

Reply: We did indeed characterize these myeloid cells and added a panel to the revised Figure 2D. It is likely that they also play a major role in the observed phenotype, since all these cell types are expelled by MYC expression.

Comment 2. *As quantifications of immune cells by flow are based on the percentage of live cells, comparison of tumor cell death in the WP, WP-MYC, WB1P, and WB1P-MYC tumor models by either staining or flow should be included to show that the decrease in immune cell populations is not due to changes in percentages of live tumor cells.*

Reply: This is a valid point in principle. We assessed H&E slides from all the tumors and could not detect major differences in the general histology of the tumors. We also performed stainings for cleaved Caspase-3 in WB1P and WB1P-Myc tumors and saw that there are more apoptotic cells in the WB1P-Myc tumors. This means that if there were a confounding effect caused by changes in percentages of live tumor cells, we would expect it to go towards more rather than fewer immune cells in the WB1P-Myc tumors. We therefore conclude that the decrease in immune cell populations in WB1P-Myc tumors is not due to changes in percentages of live tumor cells (revised Supplementary Figure S2H).

Comment 3. *Do WP and WP-MYC tumors progress at a different pace, as tumor size may impact immune cell infiltration (Kim et al, Front. Immunol., 2021 <https://doi.org/10.3389/fimmu.2020.629722>)? Were these tumors of similar sizes when collected? It would also strengthen the data to include a comparison of immune cell infiltration in tumors of similar sizes from both early and late stages of tumor development in these models.*

Reply: All tumors were harvested when they reached a size of approximately 1500mm³, so they are indeed comparable in size. They also grew at a very comparable speeds as shown in the newly added Supplementary Figure S2F. As suggested by the reviewer, we also added a comparison of immune infiltration in early-stage (10-15mm³) tumors (revised Supplementary Figure S6B). We have discussed this issue more extensively in the discussion section of the revised manuscript, and have cited the indicated reference.

Comment 4. *The authors have shown rather convincingly that MYC overexpression in tumor cells inhibits immune-cell infiltration in the TME. However, I find that the link between suppression of the STING pathway by MYC in tumor cells and its downstream paracrine effects on immune cells is not evident. Were the decreased TIL levels in MYC-overexpressing tumors due to suppression of the proinflammatory factors (such as CXCL10, CCL5, and interferons) or elevated immunosuppressive molecules secreted by the tumor cells? Can the authors measure changes in levels of cytokines and chemokines in MYC-overexpressing tumors (and/or in serum) vs in non-MYC-expressing tumors (and/or in serum)? In the in vitro co-culturing assays, would addition of some of these decreased cytokines rescue T-cell killing, T-cell migration, and T-cell proliferation in the presence of MYC-overexpressing tumor cells/organoids?*

Reply: This is a very good point raised by the reviewer. We measured the changes in cytokine and chemokine levels in culture medium from WB1P-Myc and WB1P organoids via cytokine array (revised Supplementary Figure S4C). We saw that CCL5 and CXCL10 were among the most differentially expressed cytokines and also performed ELISAs to confirm this (revised Figure 3D). Therefore, we added recombinant CCL5 and CXCL10 to organoid-CD8⁺ T cell co-cultures. We saw addition of these chemokines to the co-cultures increased the killing capacity of the CD8⁺ T cells towards WB1P-Myc organoids (revised Figure 4E). These data indicate that reduced cytokine secretion of tumor cells mediated upon MYC overexpression hampers the immune response.

Comment 5. *MYC-overexpressing tumors are normally more aggressive than non-MYC-overexpressing tumors, so it is expected that these tumors progress faster than non-MYC tumors as shown by the authors in several different mouse models. Although the authors have shown that MYC-overexpressing organoids were more resistant to T cell killing in vitro, this is not shown in vivo, and it is not clear if MYC-inhibited immune infiltration actually enhance tumor progression or simply render the tumors more resistant to immunotherapy. I wonder if the authors have further delineated these relationships. In vadimezan-treated animals in Figure 6, did authors observe an increased immune infiltration in WB1P and WB1P-MYC tumors? Do levels of immune infiltration correlate with survival or tumor growth?*

Reply: As also stated in our response to point 1 of reviewer #1, we did not observe notable differences in growth speed between the different tumor models (revised Supplementary Figure S2F). The main difference that we observed is the latency until detection of the tumors. We also present evidence for the notion that MYC-mediated immune cell expulsion confers a growth advantage for tumors in revised Figure S6D, which shows that upon de-induction of MYC through withdrawal of Tamoxifen, the tumors with the largest increase in T cell infiltrate displayed the largest growth decrease. Moreover, we quantified immune cell infiltration in tumors upon vadimezan treatment, and saw strong influx of immune cells in WB1P and WB1P-Myc tumors in the first week after treatment (revised Figure 6C,D). We also saw that at the endpoint, when tumors grew out after treatment with vadimezan was stopped, the T cell numbers were

again comparable to those observed in untreated tumors (see Figure 1 of this rebuttal). This observation further underscores the relation between T cell abundance in the tumor microenvironment with tumor growth.

Minor points:

Comment 1. Scale bars need to be added to Figures 2C, 2E, 4D, S3B, S6A, S6B, and S6D.

Reply: We thank the reviewer for spotting this omission. We added scale bars to all micrographs.

Comment 2. In Figure 2C, WB1P and WB1P-MYC are mislabeled as WP1P and WP1P-MYC.

Reply: We thank the reviewer for spotting this mistake and we remedied it.

Comment 3. Make sure the labels for figure axes are consistent. For example, in graphs for CD3 IHC quantification, the Y-axis in some figures is labelled as "CD3+ (average # of cells)", whereas others as "CD3+/area" or "CD3+ cells (ave number/area)".

Reply: We fixed the labels to be consistent.

Comment 4. In S3B, the Lenti-sgPTEN-Myc tumor image appears to be very pixelated compared with the Lenti-sgPTEN image. Do they have the same magnification?

Reply: We exchanged the images to fix the quality issues and added scale bars.

Comment 5. Why are different types of statistical tests (t-test, Mann-Whitney tests) used to determine statistical significance? The type of statistical test is not stated in figure legends for Figures 4B and 4D. Figure 3H does not have statistical analysis. Some of the statistical tests are also not used correctly. For example, in Figures 3G, 4B, 4D, 4E, and 4I, one-way or two-way ANOVA tests should be used accordingly instead of unpaired t-tests since these datasets contain multiple (>3) groups for comparison (i.e., one-way ANOVA with post hoc multiple comparisons for Figures 4E and 4I, and two-way ANOVA with post hoc multiple comparisons for Figures 3G, 4B, and 4D where two variables are included).

Reply: We now used one way ANOVA analysis, with post-hoc multiple comparisons for the analysis shown in Figure 3I and K. We used t-tests or Mann-Whitney analysis based on our assumption of normally distributed data. We have indicated per figure panel which statistical test has been used. We also included statistical analysis for Figure 3H.

For the analysis of Figures 3G, 4B, 4D and 4I, we agree that these experiments have multiple groups. However, these groups are not independent, and the experiments included a limited number of defined hypotheses which we tested. So, not all comparisons between groups are meaningful, which is why we feel ANOVA is not an appropriate test. We have used Mann-Whitney-U tests as we presume datasets are not normally distributed. We have indicated all comparisons that we have tested for statistical significance. We did use two-way ANOVA to analyze the experiments shown in Figures 4C and 4E.

Comment 6. *Is it possible to include representative images containing multiple cells and images of control cells in Figure 3F?*

Reply: We initially left out pictures containing multiple cells and images of control cells due to space limitations in the figure. We have now included images with multiple cells in revised Figure 3F.

Comment 7. *In S4F, were the shLUC transduced BT549 cells in the right panel treated with dox? If so, “+” and/or “-” symbols will need to be added. The middle panel images seem to contain protein bands cut and pasted from different blots – please clarify.*

Reply: shLuc-transduced cells were indeed treated with doxycycline. We have added the "+/-" symbols. The blots in the middle panel are indeed cut, which was inadvertently not indicated in the Figure. We have now indicated the cut-site (similar to the blots in the right panel).

Comment 8. *Survival curves of WB1P-MYC mice should also be added to Figure 6A.*

Reply: We added the Kaplan-Meier curves of the progression-free survival of WB1P-Myc mice in revised Figure 6A.

Comment 9. *In Figure 6B, perhaps a simpler and more conventional dot plots similar to Figure 6C, rather than the current waterfall plot, could be used to show changes in tumor volumes.*

Reply: We have replaced the waterfall plot with Kaplan-Meier curves in revised Figure 6A and 6B. We also added statistical analyses.

Comment 10. *In Figure 6D, the flame symbol with a cross is a bit confusing especially when it is positioned on the cell membrane. As readers may not necessarily understand a flame symbol as inflammation, perhaps include some text to explain it and connect it with the rest of the pathway? In addition, this illustration only shows what occurs within the tumor cell. How these changes impact antitumor immunity should also be demonstrated.*

Reply: We agree with the reviewer that this might be confusing. We have updated the graphical model accordingly and expanded the figure legend.

Reviewer #3:

In this paper 'MYC promotes immune-suppression in TNBC via inhibition of IFN signaling' by Zimmerli et al., the authors surveyed the TNBC samples within the TCGA database and the TONIC phase II trial and found that MYC expression is associated with IFN and inflammatory gene downregulation. Using MYC-overexpressing human TNBC cell lines and mouse TNBC models, they further showed that MYC overexpression drives immune cell tumor exclusion by suppressing expression of IFN-stimulated genes (ISGs) and inflammatory genes in a tumor-cell intrinsic manner. With CHIP-seq analysis, the authors further showed that MYC serves as a master transcriptional regulator by directly binding to the promoter regions of a myriad of IFN and inflammatory pathway genes. Overall, the study provides an important mechanism by which TNBCs and BRCA-mutated breast cancers may achieve tumor immune evasion and respond poorly to ICI. This is an interesting paper with a novel finding about the effect of MYC on tumor immunity. We appreciate the various GEMM models, organoids, and human TNBC cell lines used and the detailed bioinformatic analyses the authors performed to support their findings. However, there are few issues the authors may need to address before making a more concrete conclusion.

Major points:

Comment 1. *The authors relied on overexpressing MYC in human TNBC cell lines and mouse TNBC models to study the effect of MYC. It remains unknown what the endogenous level of MYC expression is in these models, and what is the effect of endogenous MYC on immune gene transcription and immune cell tumor recruitment. It would be more convincing to add data with MYC knockout tumor models, and to compare MYC expression in early vs late stage TNBC models and correlate endogenous MYC expression with tumor immune signature and gene expression.*

Reply: We agree with the reviewer that MYC knockout tumor models would be interesting, but such models do not exist since MYC is a common essential gene (<https://depmap.org/portal/gene/MYC>) and complete loss of MYC is detrimental for cell division and tumor growth (Wang *et al.*, *Oncogene* 2008;27:1905-15). Of note, the WB1P mouse model we used in this study develops mammary tumors with low levels of MYC expression (Annunziato *et al.*, *Nat Commun* 2019;10:397). We did check for changes in immune infiltrations in early versus late-stage tumors, but there were no striking differences (revised Supplementary Figure S6B).

To test the effects of reduced MYC expression on inflammatory signaling, we performed experiments using a doxycycline-inducible shRNA targeting MYC in BT-549 TNBC cell lines. We observed that shRNA-mediated depletion of MYC leads to increased inflammatory signaling, as measured by cytokine secretion and STAT1 phosphorylation (revised Figure 3J,K, revised supplementary Figure S5B).

Comment 2. *The authors used CHIP-seq to show that MYC functions as a master regulator by targeting numerous IFN and inflammatory signaling genes through binding to their promoter region and downregulating target gene expression. However, no mechanistic insight was provided to explain how MYC achieved this transcriptional inhibition. It would be great if the authors could offer some mechanistic evidence to support their finding or at least discuss what they are thinking.*

Reply: The reviewer brings up an important point. We found in our ChIP-seq experiments that MYC binds to many IFN and inflammatory signaling genes, in tumor organoids as well as in whole tumor extracts. The

expression of these same genes is downregulated upon MYC expression. Previous work by the Eilers lab has shown that MYC repressive function is usually mediated by complex formation with MIZ1 (Wanzel *et al.*, Trends Cell Biol 2003;13:146-50). We therefore performed ChIP-seq using MIZ1 antibody and found that MIZ1 DNA-binding overlapped with MYC ChIP-seq peaks in promoters of IFN-signaling genes, supporting our hypothesis that MYC-MIZ1 complexes transcriptionally repress these immunity genes (revised Figure 5E,F and revised Supplementary Figure S8A-E). Furthermore, we tested the ability of mammary tumor induction of a MYC version that is incapable of binding to MIZ1 (MYC-V394D, van Riggelen *et al.*, Genes Dev 2010;24:1281-94). In these experiments we saw an increase in tumor latency and immune cell infiltration upon loss of MYC-MIZ1 interaction (revised Figure 5D).

Comment 3. *Many experiments have very small sample size ($n \leq 4$), for example Figures. 3D, 4B, and 4C; or do not show statistical analysis, for example Figures. 3H, S5A, and 6B, making it hard to determine if these data support the authors' conclusions.*

Reply: We agree with the reviewer that sample sizes were small and statistics needed to be added to these figures. We therefore performed repeat experiments to increase the sample sizes and added statistical analysis (revised Figures 3J, 4B, 6A,B). We also quantified the data from the cytokine array on supernatants of WB1P and WB1P-Myc organoids (revised Supplementary Figure S4C) and conducted ELISAs for CCL5 and CXCL10 to confirm the results from the cytokine array (revised Figure 3D).

Re-analysis of the efficacy of the knock-down of BRCA1 and BRCA2 in the HCC38 and BT549 cell lines showed, that while knock-down of BRCA2 in BT-549 cells was efficient, it was variable for BRCA1 in BT-549 cells and especially variable in the HCC38 cells for BRCA1 as well as BRCA2. Due to these issues with reaching a faithful knock-down of BRCA proteins, we decided to omit the BRCA1/2 knock-down HCC38 cell line data from the revised manuscript. Using RNAi-mediated depletion of BRCA2 in human BT549 cells, we show independent validation of MYC-mediated suppression of CCL5 and phospho-STAT1 activation by ELISA and Western Blot analysis, respectively (revised Figure 3I and revised supplementary Figure S4J).

We also moved the data from the previous Figure 4C to the revised Supplementary Figure S5C, since statistical analysis of the effects of MYC on proliferation and activation of T cells in co-cultures with BT549 cells showed only a trend towards inhibition of T cell growth upon MYC expression. Accordingly, we changed our conclusions in the text.

Minor point:

Comment 1. *When analyzing numbers of tumor-infiltrating CD3+ cells in Figures. 2C and S3B, the authors did not specify the time of analysis following tumor induction. It would be more informative to isolate TILs at different time points and compare when Myc starts to affect TIL number in the tumor.*

Reply: In the mouse models shown in Figures 2C and S3B, tumor induction is mediated by the WapCre driver, which is active in mice at an age of around 3 weeks. Tumors arise 150-200 days later in the models without MYC expression, and 100 days later in the models with MYC expression. The tumors used in figures 2C and S3B were harvested and analyzed when they reached a volume of $\pm 1500 \text{mm}^3$. We have added this information to the revised manuscript on line 164.

We determined changes in T cell infiltration upon MYC induction after specified time points, as shown in revised Figure 4F-I. We also included an analysis where we compared the amount of CD3⁺ cells in early tumors at a size of $\pm 15\text{mm}^3$ (3x3mm) next to end-stage tumors at a size of $\pm 1500\text{mm}^3$ (15x15mm) (revised Supplementary Figure S6B).

In addition, it will also be important to determine whether TIL functions were impaired with Myc expression.

Reply: Impairment of TIL functions is demonstrated by (1) the ability of MYC overexpressing human tumor cells and WB1P-Myc organoids to escape splenocyte attack, and (2) the inhibition of TIL proliferation upon co-cultivation with WB1P-Myc organoids (revised Figure 4A-E).

Comment 2: *In S3B, the CD3+ cell number in WB1P-Cas9 control tumors is missing (without lentiviral injection). The data from controls will help determine whether PTEN loss on its own has already reduced CD3+ TIL numbers.*

Reply: We agree with the reviewer that this panel was missing and added the CD3⁺ T cell numbers for the WB1P-Cas9 control tumors (revised Supplementary Figure S3B).

Comment 3. *In Figures 3E and S4D, the authors analyzed pIRF3 levels in the whole tumor. It will be more informative to gate on tumor cells to determine if the effect of MYC on IRF3 phosphorylation is tumor cell-specific.*

Reply: We in fact analyzed phospho-IRF3 levels in tumor-derived organoids, so these are tumor cells only. We have clarified this in the figure legends.

Comment 4. *Figure. 6A did not show effect of vadimezan on progression-free survival of WB1P-Myc tumors.*

Reply: We added the data for the WB1P-Myc tumors in Figure 6A.

Comment 5. *In Figure. 6B, it is hard to tell the efficacy of vadimezan with this waterfall plot. It would be better to compare tumor growth kinetics and include statistical analysis.*

Reply: We replaced the waterfall plot with the more intuitive Kaplan-Meier curves (revised Figure 6A,B) and added statistical analysis.

Comment 6. *In Figure. 6C, it is expected that organoid co-culture with splenocytes should inhibit tumor growth (as shown in Figure. 4D), making MTT conversion ratios <1. It is hard to understand why the MTT conversion ratios in vehicle treated WB1P or WB1P-MYC tumors are >1.*

Reply: This effect is caused by the metabolic activity of the splenocytes in the MTT assay. We have explained this in the revised manuscript (line 403-405).

Comment 7. *Typos, Figure. 2C labels should be B1P instead of P1P. Sup Fig. S5F should be S5E.*

Reply: We want to thank the reviewer for spotting this. We fixed the labels.

Figure 1:

CD3⁺ T cells were counted in WB1P tumors upon sacrifice of mice at endpoint (tumor volume of ± 1500 mm³). There is no discernible difference in CD3⁺ infiltration levels between vehicle-treated mice and vadimezan-treated mice. Mice were treated once every 2 weeks during 6 weeks with 25mg/kg vadimezan, then treatment was stopped and tumor progression followed until reaching of endpoint.

REVIEWER COMMENTS

Reviewer #1 (Remarks to the Author):

The authors have adequately addressed most of my previous points and the manuscript has been substantially improved.

There remain some minor issues that need to be addressed.

I Shared the same concern as raised by Reviewer 2 Comment 2, which is also related to what I mentioned in my previous Comment 1. The authors' gating strategy may affect the total live cells, and therefore may impact their result.

The authors measured CCL5 and CXCL10 level using cytokine array and ELISA, but they did not elaborate how these data were normalized. Are similar number organoids plated in each analysis? Are these organoids similar in size? As such, I don't think the authors have adequately addressed the Comment 4 of Reviewer 2.

Reviewer #3 (Remarks to the Author):

The resubmission did an excellent job of answering the reviews. The paper is improved and is an important contribution.

REVIEWER COMMENTS

Reviewer #1 (Remarks to the Author):

I Shared the same concern as raised by Reviewer 2 Comment 2, which is also related to what I mentioned in my previous Comment 1. The authors' gating strategy may affect the total live cells, and therefore may impact their result.

Reply:

The comment by reviewer 2 was as follows:

Comment 2. *As quantifications of immune cells by flow are based on the percentage of live cells, comparison of tumor cell death in the WP, WP-MYC, WB1P, and WB1P-MYC tumor models by either staining or flow should be included to show that the decrease in immune cell populations is not due to changes in percentages of live tumor cells.*

To confirm that the observed decrease in immune cell populations was not due to changes in percentages of live tumor cells, we performed immunohistochemistry with an antibody against cleaved caspase-3 (cell signaling, #9661) on sections of formalin-fixed paraffin-embedded WB1P and WB1P-Myc mammary tumors. We next quantified the percentage of Caspase3-positive apoptotic cells. The results are shown in Supplementary Figure 2H of the revised manuscript and included below in this point-by-point reply (Figures 1 and 2).

We observed that the percentages of apoptotic cell rates are slightly higher in WB1P-Myc tumors than in the WB1P tumors. This means that our quantifications of immune cells by flow – which are based on the percentage of live cells – would result in a slight overestimation of the number of immune cells in WB1P-Myc tumors, compared to WB1P tumors.

This issue by no means invalidates our findings. On the contrary, it means that our FACS results slightly underestimate the MYC-mediated decrease in immune cell populations in WB1P-Myc tumors.

We have clarified this in the text of the revised manuscript (page 8, lines 192-194).

The authors measured CCL5 and CXCL10 level using cytokine array and ELISA, but they did not elaborate how these data were normalized. Are similar number organoids plated in each analysis? Are these organoids similar in size? As such, I don't think the authors have adequately addressed the Comment 4 of Reviewer 2.

Reply:

We apologize for the fact that we did not elaborate how the cytokine array and ELISA data were normalized. For these assays, we dissociated organoids to single cell suspensions and plated the same numbers of dissociated cells. Upon time of analysis after 48h, we confirmed that the numbers of cells in the plates were approximately similar. We normalized the data by putting the values of cytokine secretion of the WB1P organoids to 1 for every replicate of the experiment. We have included this information in the revised manuscript (page 25, lines 673-674 and 679-680).

Figure 1: Assessment of apoptotic cells in WB1P and WB1P-Myc tumors by immunohistochemistry to cleaved caspase-3.

Figure 2: Quantification of percentages of cleaved caspase-3 positive cells in WB1P and WB1P-Myc tumors.

REVIEWERS' COMMENTS

Reviewer #1 (Remarks to the Author):

The authors have addressed my questions and I have no further comments.